# Weakening of the AMOC and strengthening of Labrador Sea deep convection in response to external freshwater forcing

**Xinyue Wei** [1] ✉ **& Rong Zhang** [1,2] ✉

The Atlantic Meridional Overturning Circulation (AMOC) is a key player in climate. Here, we employ an ensemble of water hosing experiments to examine mechanisms of AMOC weakening and its subsequent impact on the Labrador Sea open-ocean deep convection. The subpolar AMOC decline in response to the external freshwater flux released over the southern Nordic Sea is dominated by that across the eastern subpolar North Atlantic, and the largest subpolar AMOC decline is at the relatively dense level around $\sigma_0 = 27.84\,\mathrm{kg\,m^{-3}}$. The AMOC decline is associated with subsurface cooling in the subpolar North Atlantic and the decline in the deep ocean west–east density contrast across the subpolar basin. Contrary to previous studies showing that the AMOC decline is caused by subsurface warming through the shutdown of the Labrador Sea open-ocean deep convection, our results reveal a different response, i.e., a strengthening of the Labrador Sea open-ocean deep convection, which is not a cause of the AMOC decline. The strengthening of the Labrador Sea open-ocean deep convection is mainly due to the relatively stronger freshening in the deep Labrador Sea associated with the freshening/weakening of the Iceland-Scotland Overflow, and thus reduced vertical stratification in the central Labrador Sea.

The Atlantic Meridional Overturning Circulation (AMOC), a key component of the Earth's climate system, is crucial in regulating climate by redistributing heat across ocean basins. The Labrador Sea open-ocean deep convection was often thought to play an important role in AMOC changes[1–3]. For example, in some previous water hosing experiments, the external freshwater flux is released broadly over the entire subpolar North Atlantic and directly covers the interior Labrador Sea deep convection site, causing the weakening and shutdown of the Labrador Sea open-ocean deep convection and associated subsurface warming that further drives the AMOC weakening[4,5]. However, the AMOC and the Labrador Sea open-ocean deep convection exhibit different tipping behaviors in climate models[6] and the importance of the Labrador Sea open-ocean deep convection in the AMOC has been challenged[7–16]. Previous studies have shown that the net sinking induced by the Labrador Sea open-ocean deep convection is negligible, the contribution

to the AMOC from the Labrador Sea is mainly from boundary sinking, and the impact of the open-ocean deep convection on the AMOC has to be indirect through eddy mixing with boundary properties[7,8,11–16]. Hence the AMOC (associated with the net sinking) and open-ocean deep convection (with little direct contribution to the net sinking) involve different physical processes.

Recent observations from the Overturning in the Subpolar North Atlantic Program (OSNAP) show that the AMOC across OSNAP East, rather than the AMOC across OSNAP West, is the dominant component of the subpolar AMOC[17–19]. The observational-based Robust Diagnostic Calculations also show that the Labrador Sea open-ocean deep convection contributes minimally to the long-term mean AMOC strength[20]. These findings question the pronounced role of the Labrador Sea open-ocean deep convection in the AMOC. A recent study[21] shows that the relative role of the Labrador Sea deep convection in

[1]Program in Atmospheric and Oceanic Sciences, Princeton University, Princeton, NJ, USA. [2]NOAA/OAR/GFDL, Princeton, NJ, USA.
✉e-mail: xinyuew@princeton.edu; rong.zhang@noaa.gov

multidecadal AMOC variability is model dependent: in the model with unrealistically strong/wide mean state Labrador Sea deep convection, multidecadal AMOC variability is dominated by multidecadal variability in the Labrador Sea deep convection; whereas in the model with a relatively weak/narrow mean state Labrador Sea deep convection, multidecadal AMOC variability is dominated by multidecadal Arctic salinity variability. The model with unrealistically strong/wide mean state Labrador Sea deep convection has a higher-than-observed density along the Labrador Sea boundary outflow and overestimates the mean state AMOC across OSNAP West[22–24]. The dominant role of the Labrador Sea deep convection in multidecadal AMOC variability might also be overestimated in models with overestimated mean state AMOC across OSNAP West due to modeling deficiencies in separating the boundary outflow from unrealistically strong/wide deep convection in the Labrador Sea.

The Labrador Sea Water (LSW) is formed through winter deep convective mixing, which is often driven by strong surface heat losses[25,26]. The spread of the LSW is important for ventilating the interior intermediate-depth ocean[27,28]. Beneath the LSW lies the Northeast Atlantic Deep Water (NEADW) layer, modified from the Iceland-Scotland Overflow Water (ISOW) that flows into the subpolar North Atlantic through the Iceland-Scotland Ridge from the Nordic Sea. The Iceland-Scotland overflow is an important deep branch of the AMOC[29] and can affect the westward contraction/eastward expansion of the subpolar gyre in the upper North Atlantic[30]. As the ISOW crosses the Faroe Bank Channel into the Iceland Basin, it is modified by the warm and saline water in the upper eastern subpolar North Atlantic, transporting relatively saltier water downward into the deep Labrador Sea along multiple interior downstream Iceland-Scotland overflow pathways[20,31,32] and forming the NEADW layer in the deep Labrador Sea[33]. This downward propagation/entrainment process of the upper eastern subpolar North Atlantic water properties into the ISOW is also supported by recent observations[34]. Previous studies have focused on the role of surface buoyancy loss in affecting Labrador Sea open-ocean deep convection[25,26,35], and the influence of ISOW on the Labrador Sea open-ocean deep convection changes receives less attention. In the abyssal layer of the Labrador Sea, the coldest and densest Denmark Strait Overflow Water (DSOW) resides. The DSOW flows into the subpolar North Atlantic through the Denmark Strait from the Nordic Sea and travels to the Labrador Sea's bottom layer along the western boundary of the Irminger Sea and the Labrador Sea.

In this work, we aim to explore if it is possible to have an alternative relationship between the AMOC change and the Labrador Sea open-ocean deep convection change in response to the external freshwater forcing applied over the southern Nordic Sea where there is no open-ocean deep convection (see Methods section). We employ an advanced coupled climate model that can simulate more realistic ocean boundary currents and deep overflows (see Methods section) to unravel the intricate dynamics governing the AMOC response and its relationship with the Labrador Sea open-ocean deep convection response. Our study also reveals how the external freshwater forcing applied over the southern Nordic Sea alters the density and the flow strength of the ISOW, and ultimately affects the open-ocean deep convection strength in the Labrador Sea.

## Results

### The weakening of the AMOC and the strengthening of the Labrador Sea open-ocean deep convection

With 0.05 Sv of external freshwater flux added over the southern Nordic Sea for the entire 80-year duration of the experiment (see Methods section), the maximum AMOC across the OSNAP section weakens by about 2.6 Sv (significant with 95% confidence) averaged over the last 40 years (Fig. 1a), which is -15% of the climatology maximum AMOC (17.2 Sv) averaged over the last 40 years in the corresponding control simulation. The AMOC at a relatively dense level

around $\sigma_0 = 27.84$ kg m$^{-3}$ (a level denser than the level of the maximum AMOC) has the largest and statistically significant weakening of 5.7 Sv averaged over the last 40 years (Figs. 1a, 2c). The rapid decline occurs in the first 40 years and the weakened AMOC is close to an equilibrium after 40 years (Fig. 1a). It takes a longer time for the weakened AMOC to reach an equilibrium at the relatively dense level.

The AMOC declines across a broad range of latitudes (extending from the subpolar region to the low latitudes) and the decline is larger across subpolar latitudes than that across low latitudes (Supplementary Fig. 1). The AMOC weakening across the OSNAP section is dominated by that across OSNAP East, with minimal contribution from the OSNAP West component (Figs. 1b, 2), though the control simulation overestimates the density-space AMOC strength across OSNAP West (see Methods section). This result is consistent with recent OSNAP observations[17,19] that the AMOC across OSNAP East dominates the AMOC across the entire section. In contrast to previous studies attributing the AMOC weakening to the shutdown of the Labrador Sea open-ocean deep convection and associated subpolar subsurface warming[4,5], our study reveals a distinct AMOC weakening mechanism that does not coincide with the reduction of the Labrador Sea open-ocean deep convection (Fig. 1a, c, d, e). Here the AMOC weakening is associated with the subsurface cooling (Fig. 3e, Supplementary Fig. 2), rather than the subsurface warming over the subpolar North Atlantic as found in previous water hosing experiments[4,5]. Our results are also consistent with observational analyses suggesting that the inferred historical weak AMOC phase is associated with the time-lagged subsurface cooling over the central Labrador Sea and vice versa[36].

The central Labrador Sea March mixed layer depth (MLD) does not decline in the first 40 years and even has a statistically significant increase averaged over the last 40 years (Fig. 1c, e), indicating that the Labrador Sea open-ocean winter deep convection strengthens, though the AMOC declines in response to the external freshwater forcing. The strengthening of the winter deep convection spreads over the open ocean in the Labrador Sea and Irminger Sea (Fig. 1e). Thus, the AMOC weakening (Figs. 1a, 2c) is opposite to the strengthening of the Labrador Sea open-ocean deep convection (Fig. 1c, e). Even the modest reduction of the AMOC across the Labrador Sea (OSNAP West, Figs. 1b, 2a) is also opposite to the strengthening of the Labrador Sea open-ocean deep convection (Fig. 1c, e). These results are consistent with previous analyses showing that the AMOC (associated with the net sinking) and open-ocean deep convection (with little direct contribution to the net sinking) involve different physical processes[7,8,11–16].

### The role of water mass transformation in the AMOC weakening

The Water mass transformation (WMT) refers to the diabatic processes by which water masses transform from one density class to another, influencing/balancing the large-scale ocean circulation such as the AMOC[37]. The WMT includes the surface forced (WMT$_S$) and the interior mixing forced (WMT$_M$) components (see Methods section for the calculation methods). We investigate how the long-term mean surface and interior mixing forced water mass transformation (WMT$_S$ and WMT$_M$) changes influence/balance the long-term mean density-space AMOC decline. Averaged over the last 40 years, the AMOC decline across the Greenland-Scotland Ridge (GSR) (Supplementary Fig. 3c) is affected by both WMT$_S$ and WMT$_M$ anomalies northeast of GSR that often counter each other at various density levels (Supplementary Fig. 3i). In the Iceland-Irminger Seas (IIS), the total WMT anomaly is mainly dominated by the WMT$_M$ anomaly, with little change in the WMT$_S$ anomaly (Supplementary Fig. 3b, h). The AMOC decline across OSNAP East (Supplementary Fig. 3a) is mainly balanced by the WMT$_S$ decline occurring in large part remotely northeast of GSR at the relatively dense level around 27.84 kg m$^{-3}$ (Supplementary Fig. 3a, g), because the WMT$_M$ anomalies for the IIS region and the region northeast of GSR are almost canceled at this density level. The above long-term mean balance does not necessarily apply to the transient

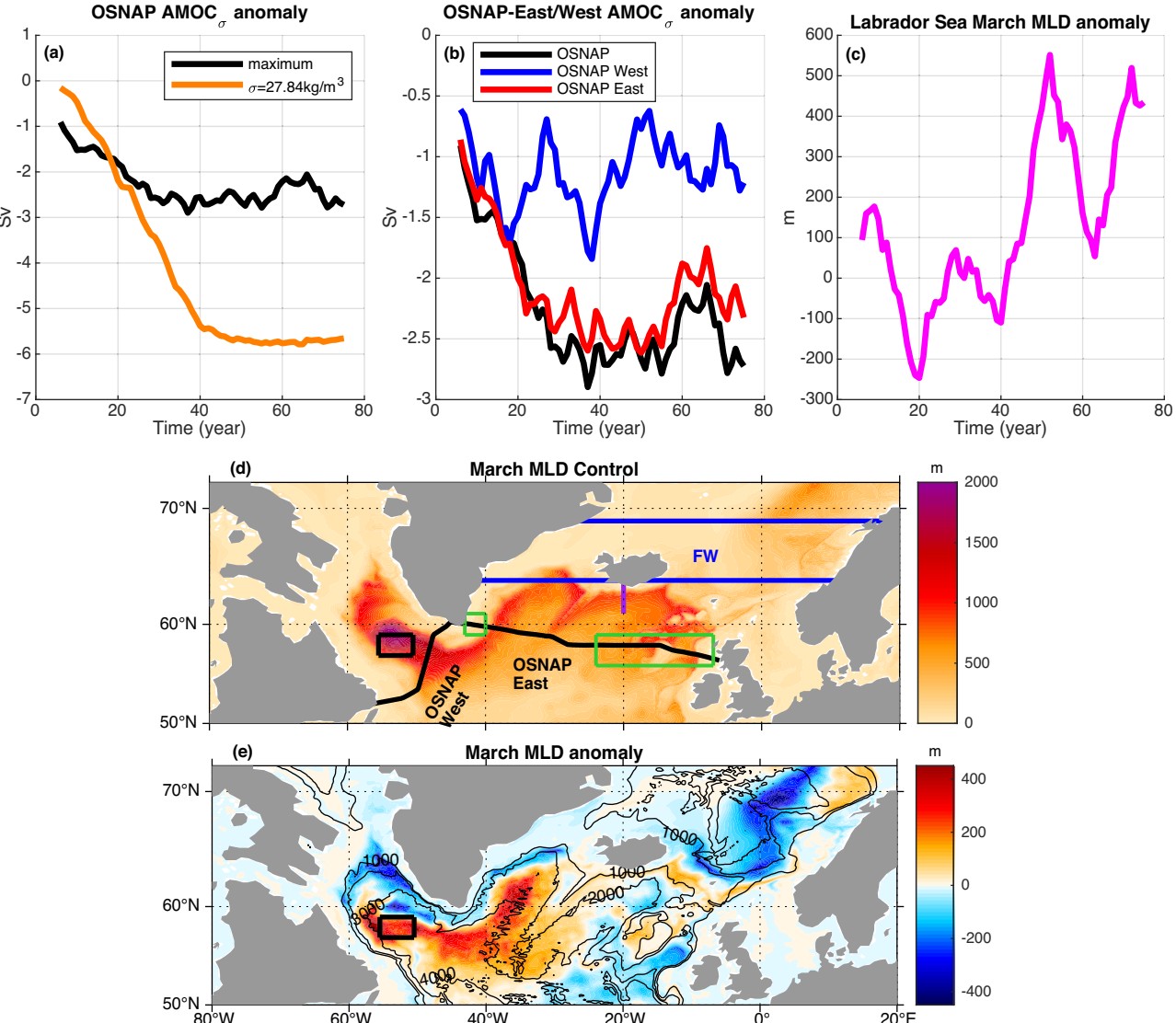

**Fig. 1 | The anomalies of the Atlantic Meridional Overturning Circulation (AMOC, Sv, 1Sv = 10⁶ m³ s⁻¹) across the Overturning in the Subpolar North Atlantic Program (OSNAP) section and the Labrador Sea March mixed layer depth (MLD, m). a** Time series of the anomalies of the maximum AMOC (black line) and the AMOC at a relatively dense level around $\sigma_0 = 27.84$ kg m⁻³ (orange line) across the entire OSNAP section in density space. **b** Time series of the anomalies of the maximum AMOC across the entire OSNAP section (black line), OSNAP West (blue line), and OSANP East (red line) in density space. **c** Time series of the March MLD in the Labrador Sea. The 11-year running mean is used in (**a**–**c**). The color maps of March MLD climatology (in the control simulation) (**d**) and anomaly (water

hosing - control) (**e**) are averaged over the last 40 years. Blue lines mark the northern and southern boundaries of the water hosing region over the southern Nordic Sea. Black line shows the OSNAP section. Purple line shows the meridional section along the Iceland-Scotland overflow pathway used in Figs. 8, 9. Black box marks the location of the open-ocean deep convection region in the central Labrador Sea (similar to the location of that observed[8,67]), which is used in (**c**) and Supplementary Fig. 8. Green boxes cover the west and east regions around the OSNAP East subsection used in Fig. 7. Bathymetric contours are shown in (**e**). The MLD is defined with a density criterion of 0.03 kg m⁻³. The open-ocean deep convection in the Labrador Sea intensifies (**c**) as the AMOC weakens (**a**).

evolution: at the relatively dense level, the WMT_S decline northeast of GSR reaches equilibrium much earlier around 20 years, whereas the AMOC decline across OSNAP East in the corresponding deep ocean has a much longer decline timescale of more than 40 years (discussed in a later subsection).

### The horizontal circulation contribution to the AMOC weakening across OSNAP East

In the control simulation, the climatological maximum AMOC in density space is much larger than that in depth space across OSNAP East (Fig. 2b, e) due to the contribution from the cyclonic horizontal circulation across sloping isopycnals, which can be estimated from the difference between the density-space and depth-space AMOC[20]. Across OSNAP East, the largest AMOC reduction is at a relatively dense

level around $\sigma_0 = 27.84$ kg m⁻³ (Fig. 2b, which on average is around 2000m). At this relatively dense level, the depth-space AMOC has little reduction (i.e. 0.8 Sv around 2000m) (Fig. 2e); instead, the reduction of the horizontal circulation contribution (4.9 Sv) dominates the total density-space AMOC reduction of 5.7 Sv (Fig. 2b, e).

To understand the horizontal circulation contribution to the AMOC weakening across OSNAP East, we calculate the $\sigma$–z diagram of the AMOC transport across OSNAP East (Fig. 4) based upon the AMOC diagnostics developed by Zhang and Thomas[20]. The $\sigma$–z diagram for the control simulation (Fig. 4a) resembles the observational-based $\sigma$-z diagram across OSNAP East[20]. In the climatological mean state, the cyclonic abyssal circulation in the deep North Atlantic[38] across sloping isopycnals includes the relatively denser southward western boundary current and the relatively lighter northward current near the eastern

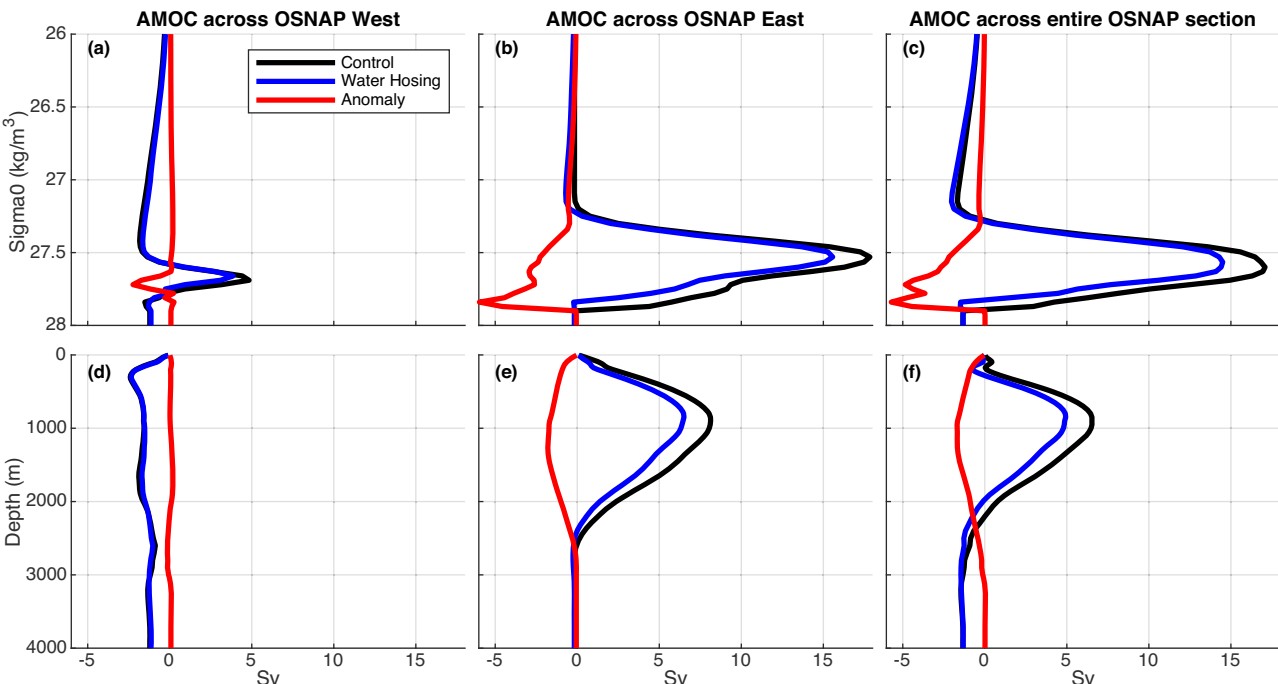

**Fig. 2 | The climatological mean and anomalies of Atlantic Meridional Overturning Circulation (AMOC) streamfunction (Sv, 1Sv = 10⁶ m³ s⁻¹) across the Overturning in the Subpolar North Atlantic Program (OSNAP) section.** a–c Density space (potential density $\sigma_0$, kg m⁻³). d–f Depth space (m). a, d OSNAP West. b, e OSNAP East. c, f Entire OSNAP section. Black lines represent the control simulation. Blue lines represent the water hosing experiment. Red lines represent anomalies (water hosing−control). The last 40-year average is used.

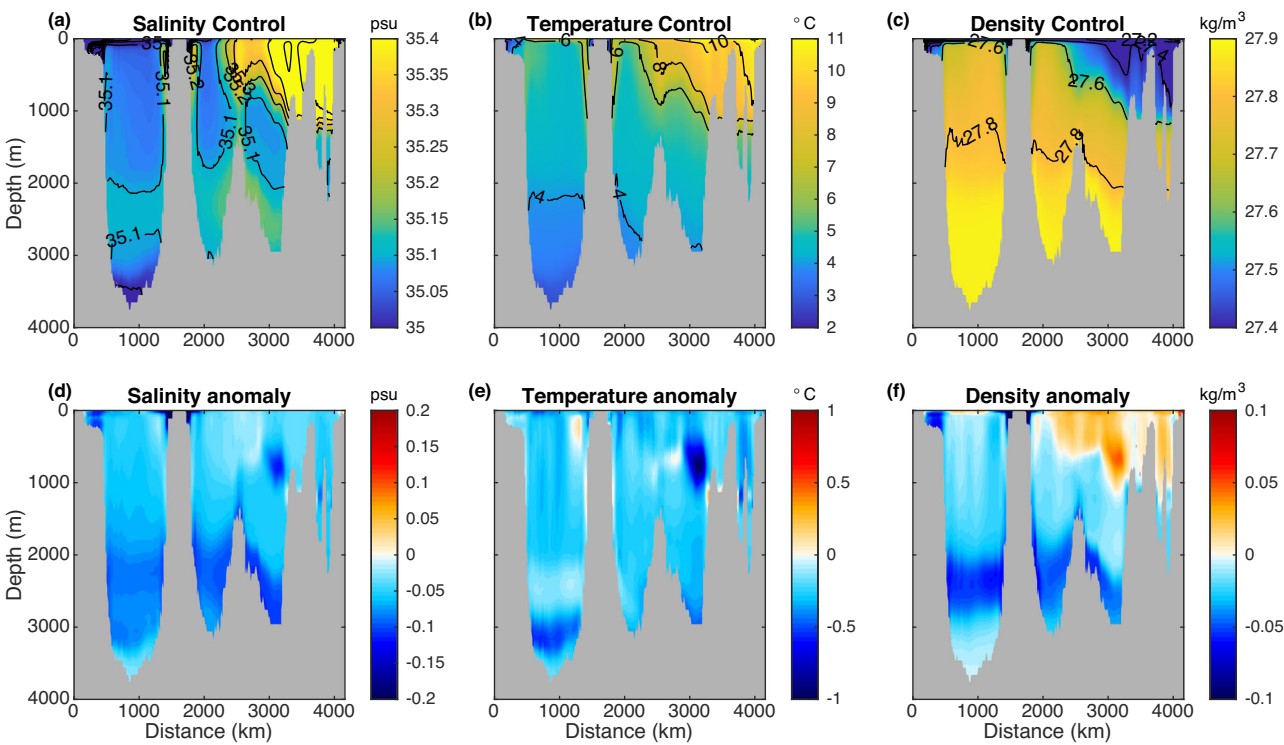

**Fig. 3 | The climatological mean and anomalies of salinity (psu), potential temperature (°C), and potential density (kg m⁻³) across the Overturning in the Subpolar North Atlantic Program (OSNAP) section.** a–c Climatology in the control simulation. d–f Anomalies (water hosing−control). a, d Salinity. b, e Potential temperature. c, f Potential density ($\sigma_0$). The last 40-year average is used.

boundary at the same depth in the deep ocean across OSNAP East (Supplementary Fig. 4). It contributes substantially to the density-space AMOC at the relatively dense level (Fig. 2b, e; Fig. 4a), an observed feature that is often missing in climate models[20]. In response to the external freshwater forcing, the largest volume transport reduction happens at the relatively dense level (Fig. 4c). However, the volume transport change is almost canceled at the deep depth level and thus there is little net depth-space volume transport change in

the deep ocean summed over the entire density range. The opposite volume transport changes at the same depth level but different density levels are related to the flattening of isopycnals (i.e.,

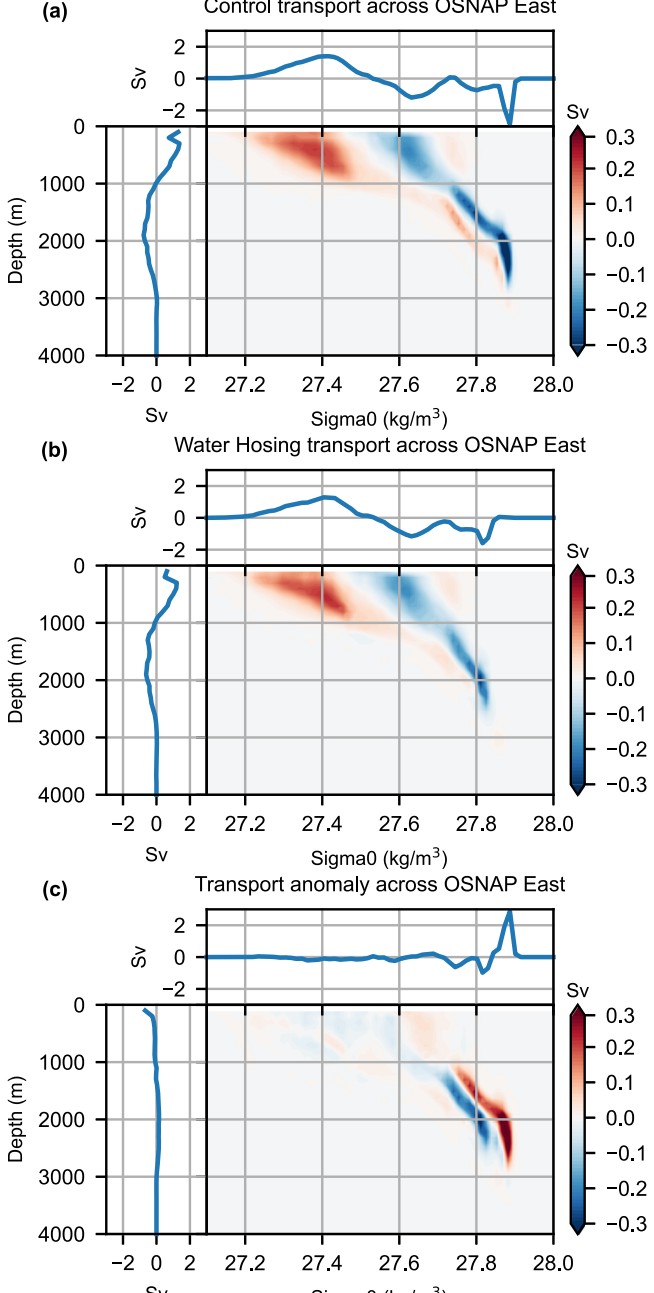

**Fig. 4 | The $\sigma$–$z$ diagram of climatological mean and anomalies of Atlantic Meridional Overturning Circulation (AMOC) transport (Sv) across Overturning in the Subpolar North Atlantic Program (OSNAP) East. a** Control simulation. **b** Water hosing experiment. **c** Anomaly (water hosing−control). In (**a**, **b**), the shaded color represents the integrated volume transport (Sv) across OSNAP East over each potential density ($\sigma_0$) bin (x-axis) and depth (z) bin (y-axis). The blue curve above illustrates the sum of the volume transport over the full depth range at each potential density bin. The blue curve on the left side shows the sum of the volume transport over the entire range of potential density at each depth bin. The accumulated AMOC transport in density- and depth- space corresponds to the AMOC streamfunction across OSNAP East, as depicted in Fig. 2b, e. The $\sigma$–$z$ diagram of AMOC transport (Sv) and its interpretation is adapted from Zhang and Thomas[20]. The last 40-year average is shown. The reduction of the horizontal circulation contribution dominates the reduction of the AMOC across OSNAP East at the relatively dense level (**c**).

reduction of the west-east density contrast) in the deep ocean (Figs. 3c, f, 4c). Therefore, the $\sigma$-$z$ diagram further supports our findings that the reduction of the horizontal circulation contribution dominates the reduction of the AMOC across OSNAP East at the relatively dense level.

### The transient AMOC decline across OSNAP East

In this subsection, we focus on understanding the transient AMOC decline in response to the external freshwater forcing. Our analyses show that the AMOC weakening across the OSNAP section is dominated by that across OSNAP East at the relatively dense level around $\sigma_0 = 27.84\,\mathrm{kg\,m^{-3}}$ (Fig. 1a, b), which is affected by the deep ocean density contrast across this subsection (Figs. 3, 5, Supplementary Fig. 5). The deep ocean AMOC decline timescale across OSNAP East is consistent with the decline timescale of the deep ocean west−east density contrast across the subsection. Both the deep ocean west−east density contrast and the AMOC at a relatively dense level around $\sigma_0 = 27.84\,\mathrm{kg\,m^{-3}}$ decline rapidly within the first 40 years, then decrease slowly and reach equilibrium around year 60 (Fig. 5c).

The density decreases in the western boundary of OSNAP East, and it is dominated by the freshening (Figs. 3, 5, Supplementary Fig. 5). The externally forced freshwater anomaly in the southern Nordic Sea suppresses upper ocean mixing and induces a cold anomaly there, which spreads into the North Atlantic along with the freshwater anomaly and provides a density compensation to the freshening (Supplementary Fig. 5). Additionally, the initial AMOC weakening induces enduring cooling in the upper eastern subpolar North Atlantic (Supplementary Fig. 2) due to reduced poleward ocean heat transport and the eastward shift of the North Atlantic current (NAC) pathway as found in previous studies[39–44]. Part of the cold anomaly is entrained into the deep eastern subpolar North Atlantic (Supplementary Fig. 5). In the deep ocean, the density decline in the western boundary is dominated by the freshening (Fig. 5a), whereas the eastern part experiences a nearly full density compensation between cooling and freshening, resulting in a negligible density change (Fig. 5b). Despite the dominance of freshening in the deep western boundary density decline (Fig. 5a), the freshening in the eastern side gradually develops in the deep ocean (Fig. 5b), so the contribution of freshening to the overall decline in the deep ocean west−east density contrast is relatively smaller after ~35 years (Fig. 5c). Instead, it is the more pronounced and enduring cooling in the eastern part compared to the western boundary (red lines in Fig. 5a, b) that primarily contributes to the decline in the deep ocean west−east density contrast after ~35 years (Fig. 5c).

### The evolution of salt-based and dye-based freshwater fraction anomalies

To understand the transient oceanic response to the external freshwater forcing, it is important to understand the evolution of freshwater fraction anomalies. With the passive tracer (dye) added at the same rate and location over the southern Nordic Sea as the external freshwater flux, the propagation paths of the externally forced freshwater anomaly can be revealed by the dye-based freshwater fraction anomaly ($FWF_{dye}$), which is represented by the dye concentration (Figs. 6, 7, see Methods section).

Although substantial dye (externally forced freshwater anomaly) is retained in the upper Nordic Sea by the local gyre recirculation, some moves out of the southern Nordic Sea and reaches the western boundary of the OSNAP East subsection through the Denmark Strait (Figs. 3d, 6, 7, Supplementary Fig. 6). The upper ocean dye propagates along the western boundary of the Irminger and Labrador seas with partial sinking/mixing into the deeper ocean, and is partially mixed into the interior ocean including the Labrador Sea open-ocean deep convection region (Figs. 6, 7, Supplementary Fig. 6). Some surface dye also gets into the relatively deeper layer in the eastern Nordic Sea through surface dense water formation there, recirculates

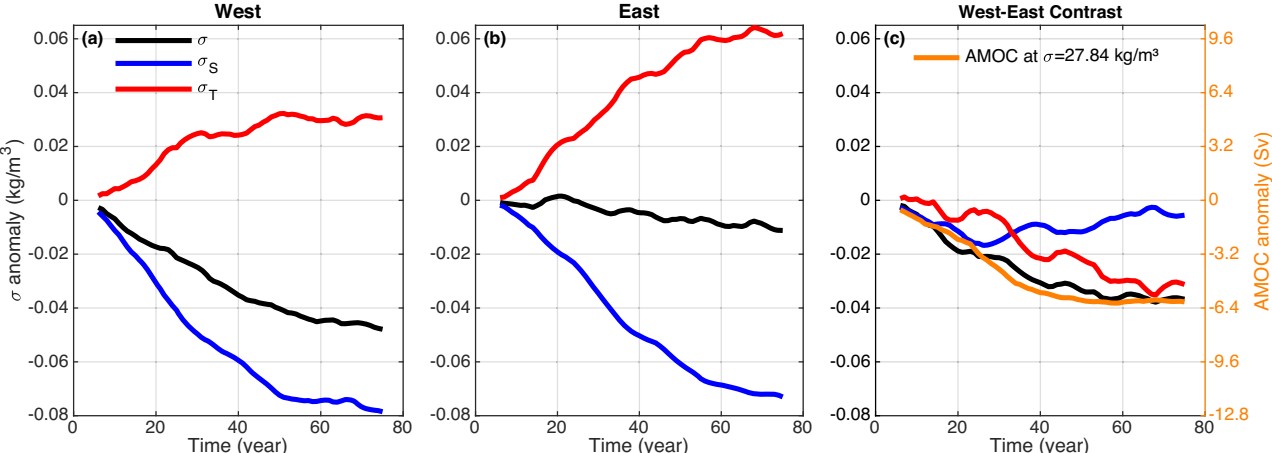

**Fig. 5 | Time series of deep ocean potential density anomalies along with its thermal and haline components over the west boundary and eastern regions of the Overturning in the Subpolar North Atlantic Program (OSNAP) East subsection.** The anomalies of potential density (black line), haline component (blue line), and thermal component (red line) averaged over the west boundary region (**a**), eastern region (**b**) of the OSNAP East subsection, and their differences (west–east contrast) (**c**), averaged over the deep ocean (2000–3000 m), with the potential density referenced to 2500 m. The west boundary and eastern regions of the OSNAP East subsection are shown as the green boxes in Fig. 1d. The orange line in (**c**) represents the Atlantic Meridional Overturning Circulation (AMOC) anomalies at a relatively dense level around $\sigma_0 = 27.84$ kg m$^{-3}$ across OSNAP East, which are consistent with the evolution of the anomalies of the deep ocean averaged (2000–3000 m) west–east potential density contrast across the subsection (black line in **c**). The 11-year running mean is used.

in the Nordic Sea, and moves out of the Nordic Sea with the Denmark Strait and Iceland-Scotland overflows. The descending Nordic Sea overflows carry the dye at the relatively deeper layer of the Greenland-Scotland Ridge directly into the deep subpolar North Atlantic along the overflow pathways (Supplementary Fig. 6), and the dye gradually spreads into the interior deep North Atlantic (Fig. 6). Both the horizontal maps over the extra-tropical North Atlantic and the vertical section of the dye-based freshwater fraction anomaly ($FWF'_{dye}$) along the OSNAP section demonstrate that the dye propagates downward from the upper ocean into the deep ocean (Figs. 6, 7, Supplementary Figs. 6, 7b).

The dye-based FWF anomaly $FWF'_{dye}$ only reflects the direct contribution from the external freshwater forcing, i.e., the advection/diffusion of the externally forced freshwater anomaly. Meanwhile, the salt-based FWF anomaly $FWF'_{salt}$ (Figs. 6a–e, 7a–e, see Methods section) reflects the total FWF anomaly. Hence the difference between the salt-based and the dye-based FWF anomalies ($FWF'_{salt} - FWF'_{dye}$) (Figs. 6k–o, 7k–o) reflects the additional FWF anomaly induced by changes in ocean circulation and mixing acting on the climatological FWF and the coupled feedback through the advection/diffusion of the additional FWF anomaly.

One of the main differences ($FWF'_{salt} - FWF'_{dye}$) happens along the NAC pathway in the upper North Atlantic (Fig. 6k–o), where there is a pronounced salt-based FWF anomaly ($FWF'_{salt}$) but little dye-based FWF anomaly ($FWF'_{dye}$) (Fig. 6). Their difference indicates that the enhanced freshening along the NAC pathway is not due to the direct advection/diffusion of the externally forced freshwater anomaly, but is induced indirectly through the AMOC weakening and associated reduced ocean salt transport/eastward shift of the NAC pathway as found in previous studies[40,45–47]. Another difference between the salt-based and the dye-based FWF anomalies is along the Iceland-Scotland overflow pathway (Fig. 7k–o), which will be discussed later in the context of the freshening and slowdown of the Iceland-Scotland overflow.

The freshening along the NAC pathway in the upper North Atlantic becomes stable after about 30 years (Fig. 6a–e). However, the deep North Atlantic keeps freshening for up to about 50 years (Fig. 7a–e). This is consistent with the longer timescale of the AMOC decline in the relatively dense deep ocean (Fig. 1a).

## The role of ISOW in the Labrador Sea open-ocean deep convection

A surprising response to the external freshwater forcing is the strengthening of the Labrador Sea open-ocean deep convection, which is related to the reduction of the vertical stratification in the central Labrador Sea as reflected in the increase of the difference in density anomalies between the surface and the deep central Labrador Sea after 40 years (Supplementary Fig. 8). The vertical density difference anomaly and the MLD anomaly in the central Labrador Sea exhibit very similar time evolution (Supplementary Fig. 8). A stronger density reduction in the deep central Labrador Sea than that in the layer above makes the central Labrador Sea less stratified and more prone to deep convection in the last 40 years (Supplementary Fig. 8; Fig. 3f).

This deep density reduction is primarily attributed to the pronounced freshening in the NEADW layer (Fig. 3d; Supplementary Fig. 7a), a layer of water that originates from the Iceland-Scotland overflow and has the maximum climatological mean state salinity (Fig. 3a). The freshening in the NEADW layer in the deep central Labrador Sea is stronger than that in the upper central Labrador Sea (Fig. 3d, Supplementary Fig. 7a).

As an indicator of the direct propagation of the freshwater anomaly, the dye-based FWF anomaly ($FWF'_{dye}$) peaks at the surface and reduces with depth in the Labrador Sea (Supplementary Fig. 7b). In contrast, the salt-based FWF anomaly ($FWF'_{salt}$) peaks in the deep Labrador Sea (Supplementary Fig. 7a). This difference in the vertical distribution between the salt-based and the dye-based FWF anomalies ($FWF'_{salt} - FWF'_{dye}$) suggests that the enhanced freshening in the NEADW layer in the deep Labrador Sea is not solely a consequence of a direct response to the external freshwater forcing (Supplementary Fig. 7c). The impacts of changes in the ocean circulation on the climatological FWF and the advection/diffusion of the indirectly induced FWF anomaly also play a crucial role on the deep Labrador Sea freshening below 2000m (Supplementary Fig. 7c). This inference is also supported by the difference between the salt-based and the dye-based FWF anomalies along the Iceland-Scotland overflow pathway on the eastern flanks of the Reykjanes Ridge, where the salt-based FWF anomaly ($FWF'_{salt}$) is much higher than the dye-based FWF anomaly ($FWF'_{dye}$) (Fig. 7k–o). This difference further spreads into the western flanks of the Reykjanes Ridge and the western deep North Atlantic

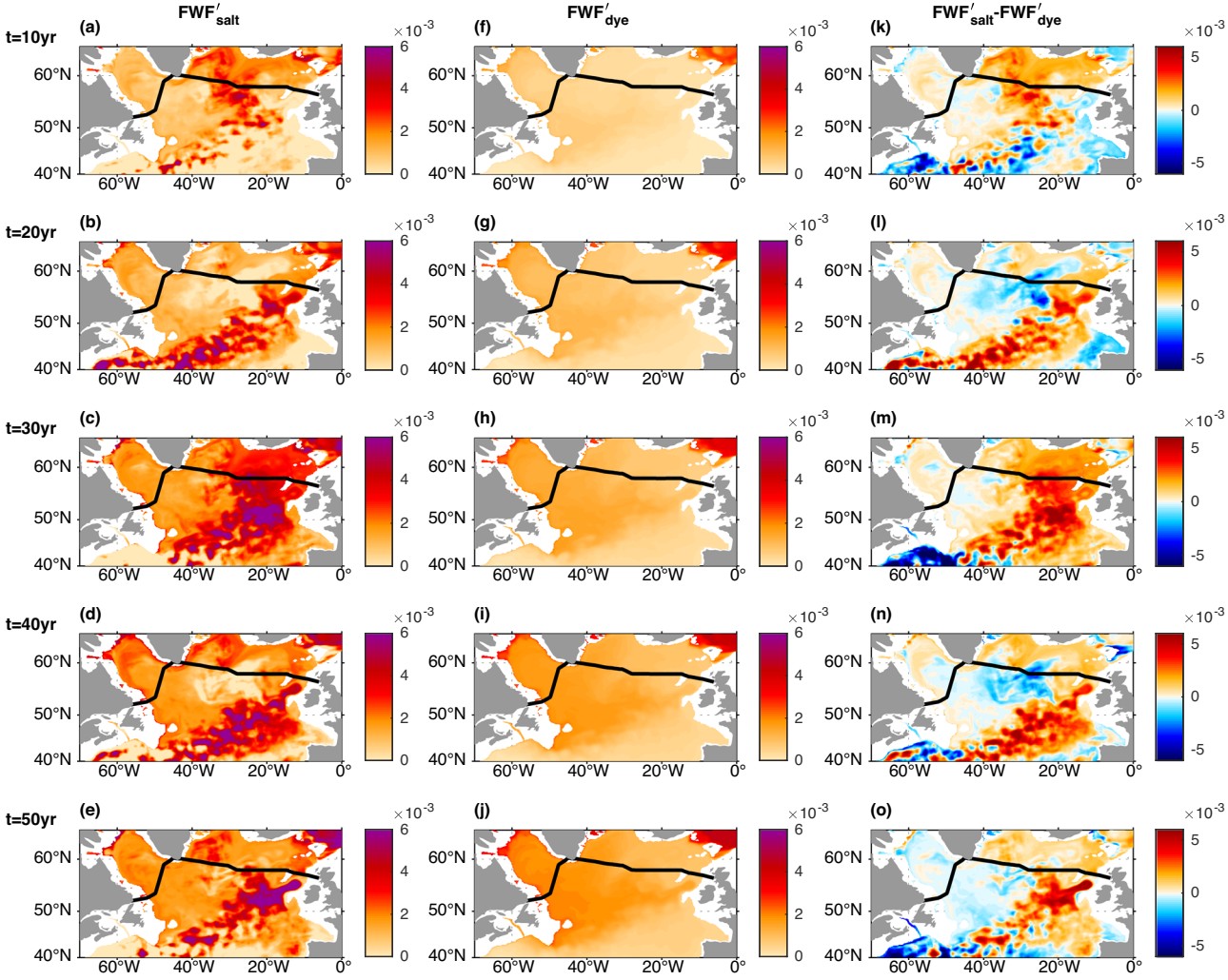

**Fig. 6 | Comparison of the transient salt-based freshwater fraction (FWF) anomalies ($FWF'_{salt}$) and dye-based FWF anomalies ($FWF'_{dye}$) at the upper ocean (413 m).** The horizontal maps show $FWF'_{salt}$ (**a**–**e**), $FWF'_{dye}$ (**f**–**j**), and their differences $FWF'_{salt} - FWF'_{dye}$ (**k**–**o**) at the upper ocean (413 m) at different years. **a**, **f**, **k** At year 10. **b**, **g**, **l** At year 20. **c**, **h**, **m** At year 30. **d**, **i**, **n** At year 40. **e**, **j**, **o** At year 50. Black lines mark the Overturning in the Subpolar North Atlantic Program (OSNAP) section.

including the Irminger Sea and the Labrador Sea along the downstream Iceland-Scotland overflow pathways (Fig. 7k–o).

To investigate changes along the Iceland-Scotland overflow pathway, we analyze a meridional section south of Iceland across the Iceland-Scotland overflow, where the climatological mean state zonal overflow in the bottom layer is westward (Fig. 8d). Along the Iceland-Scotland overflow pathway, there is a reduction in salinity (Fig. 8e), temperature (Fig. 8f), and density (Fig. 8g), and a weakening of the westward Iceland-Scotland overflow velocity (Fig. 8h). The dye-based FWF anomaly ($FWF'_{dye}$) across the Iceland-Scotland overflow pathway suggests that some externally forced freshwater anomaly is directly advected/diffused into this region (Fig. 9b). However, in contrast to the relatively uniform distribution of the dye-based FWF anomaly ($FWF'_{dye}$) below 800 m (Fig. 9b), the salt-based FWF anomaly ($FWF'_{salt}$) shows a distinct peak at the bottom, i.e. the ISOW layer (Fig. 9a). The differences between the salt-based and the dye-based FWF anomalies again indicate that the freshening is partially induced by ocean circulation changes acting on the climatological FWF and the advection/diffusion of the indirectly induced FWF anomaly (Fig. 9c).

As discussed earlier, the AMOC weakening induces the freshening in the upper eastern subpolar North Atlantic along the NAC pathway (Fig. 6k–o), which is entrained into the Iceland-Scotland overflow and contributes to the freshening of the ISOW (Fig. 7k–o). This downward

propagation/entrainment process of the upper eastern subpolar North Atlantic salinity anomaly into the ISOW is also supported by recent observations[34]. Beyond the impact from the upper eastern subpolar North Atlantic, the weakening of the Iceland-Scotland overflow also contributes to the freshening of the ISOW. The Iceland-Scotland overflow across the ISOW section (shown in Fig. 8) has a mean state volume transport (defined as the integrated volume transport over $\sigma_0 > 27.8 \, kg \, m^{-3}$) of about 3.2 Sv (similar to that observed[48]), and it exhibits almost a 50% reduction in the water hosing experiment. As the climatological mean state-modified ISOW is characteristically salty, the weakening of the Iceland-Scotland overflow reduces the salt transport to its downstream, contributing to the freshening along the downstream ISOW pathways (Fig. 7a–e). The freshening along the ISOW pathway reduces the density contrast across the ISOW section, which decreases the strength of the Iceland-Scotland overflow. This coupled positive feedback amplifies the freshening along the downstream ISOW pathway.

The climatological mean state ISOW is modified along its pathway by mixing with the warm and saline water in the upper eastern subpolar North Atlantic and becomes saltier than the original ISOW across the Iceland-Scotland Ridge[33]. As the modified ISOW flows southward along the eastern flanks of the Reykjanes Ridge into the western North Atlantic and reaches the deep Labrador Sea[33] along interior

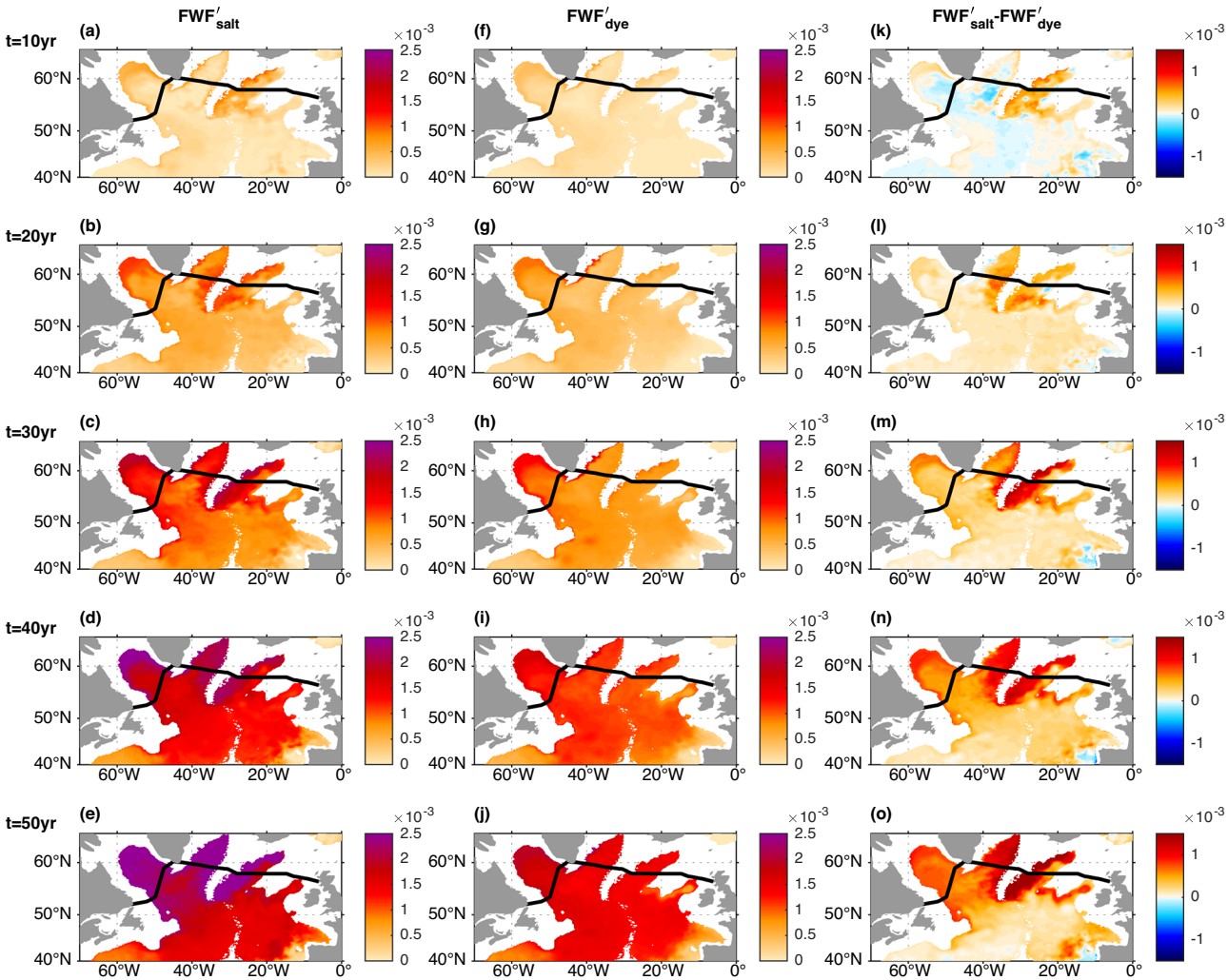

**Fig. 7 | Comparison of the transient salt-based freshwater fraction (FWF) anomalies ($FWF'_{salt}$) and dye-based FWF anomalies ($FWF'_{dye}$) at the deep ocean (2250 m).** The horizontal maps show $FWF'_{salt}$ (**a–e**), $FWF'_{dye}$ (**f–j**), and their differences $FWF'_{salt} - FWF'_{dye}$ (**k–o**) at the deep ocean (2250 m) at different years. **a, f, k** At year 10. **b, g, l** At year 20. **c, h, m** At year 30. **d, i, n** At year 40. **e, j, o** At year 50. Black lines mark the Overturning in the Subpolar North Atlantic Program (OSNAP) section.

pathways[20,31,32], it forms the Labrador Sea NEADW layer. Therefore, the freshening of the ISOW contributes to the freshening of the NEADW layer in the Labrador Sea. Moreover, because the climatological mean state modified ISOW is saltier than the water mass in the Labrador Sea, the weakening of the modified ISOW inflow strength also freshens the NEADW layer in the Labrador Sea. Combined with the direct downward mixing of the externally forced freshwater anomaly from the upper Labrador Sea, the NEADW layer becomes the most freshened layer in the central Labrador Sea (Fig. 3d).

In summary, the freshening in the NEADW layer in the deep Labrador Sea can be attributed to three primary sources: (1) the direct ocean advection/diffusion of the externally forced freshwater anomaly from the Nordic Sea into the deep Labrador Sea, (2) a decreased transport of the climatological mean state salty water into the deep Labrador Sea due to the weakened Iceland-Scotland overflow strength, and (3) the downward advection/diffusion of the upper eastern subpolar North Atlantic freshening (indirectly induced by the AMOC weakening) along the Iceland-Scotland overflow pathway, contributing to the freshening of the ISOW and the associated NEADW in the Labrador Sea. As the density reduction in the upper central Labrador Sea is less than that in the deep NEADW layer after 40 years (Fig. 3f), the central Labrador Sea becomes less vertically stratified. As a result, the

Labrador Sea open-ocean deep convection is strengthened after 40 years (Supplementary Fig. 8).

In contrast to the central Labrador Sea, the March MLD is reduced over the northern boundary of the Labrador and Irminger Seas and the southeastern boundary of the Nordic Sea (Fig. 1e). The climatological winter convection is often much shallower (Fig. 1d) in these regions mainly over continental slopes. The upper ocean density decline induced by the freshening along the upper ocean boundary currents, as seen in both salt-based and dye-based FWF anomalies (Fig. 6), dominates the reduction of March MLD over these continental slope regions.

## Discussion

Our study shows that the AMOC across the OSNAP section weakens when an external freshwater flux is added over the southern Nordic Sea where there is no open-ocean deep convection, and the AMOC weakening is dominated by that across OSNAP East, not by that across OSNAP West. The largest AMOC weakening across the OSNAP section is at the relatively dense level around $\sigma_0 = 27.84\ \text{kg m}^{-3}$ and consistent with the decline in the deep ocean west–east density contrast across the basin, but inconsistent with the changes in the Labrador Sea open-ocean deep convection. This finding contrasts with previous studies

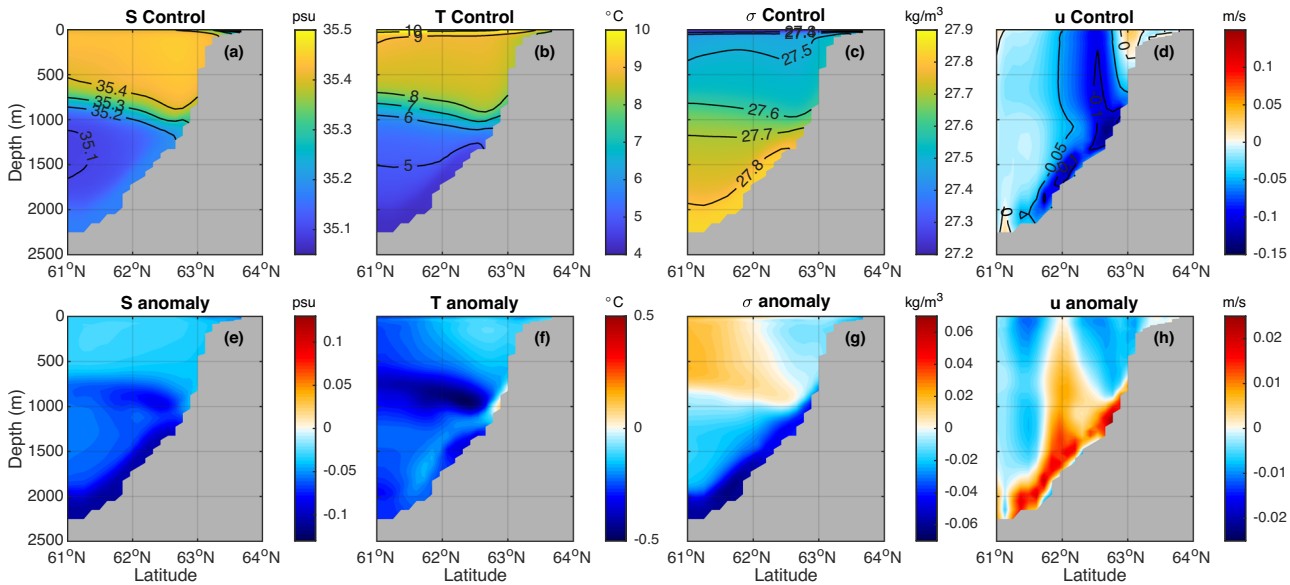

**Fig. 8 | Climatological mean and anomalies of water properties and zonal velocity along the Iceland-Scotland Overflow pathway.** A meridional section across the Iceland-Scotland Overflow pathway (purple line in Fig. 1d) of salinity (psu; **a**, **e**), potential temperature (°C; **b**, **f**), potential density $\sigma_0$ (kg m⁻³; **c**, **g**) and zonal velocity (m s⁻¹, positive to the east; **d**, **h**). **a**–**d** Control simulation. **e**–**h** Anomalies (water hosing−control). The last 40-year average is used.

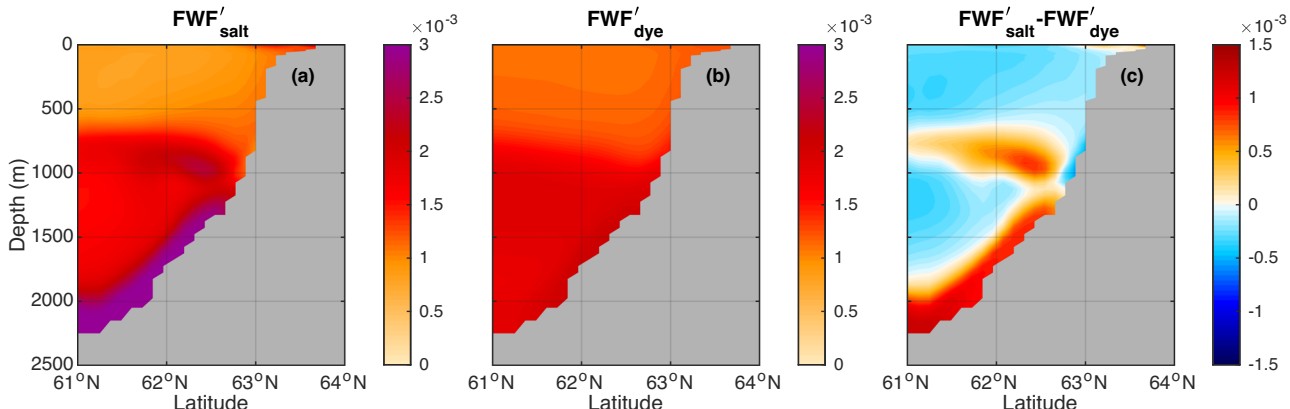

**Fig. 9 | Comparison of salt-based freshwater fraction (FWF) anomalies ($FWF'_{salt}$) and dye-based FWF anomalies ($FWF'_{dye}$) across the Iceland-Scotland Overflow section. a** $FWF'_{salt}$. **b** $FWF'_{dye}$. **c** $FWF'_{salt} − FWF'_{dye}$. The meridional section is the same as used in Fig. 8. The stronger freshening along the Iceland-Scotland Overflow pathway is partially induced by the weakening of the Iceland-Scotland Overflow. The last 40-year average is used.

where the prescribed external freshwater forcing often covers the Labrador Sea open-ocean deep convection region directly and the Labrador Sea open-ocean deep convection shuts down, causing the subsurface warming and the AMOC weakening[4,5]. In contrast, here the external freshwater forcing does not cover the Labrador Sea directly, and the AMOC weakening is associated with cooling rather than warming over the subsurface subpolar North Atlantic. The largest AMOC weakening occurs at the relatively dense level, mainly due to the weakening of the horizontal circulation contribution across sloping isopycnals. Its decline timescale is consistent with the decline timescale of the west–east density contrast in the deep ocean. The AMOC decline is a dynamic process coupling west–east density contrast changes with ocean circulation changes, and it gradually reaches the equilibrium state of the coupled dynamic system.

Our study also shows that the Labrador Sea open-ocean deep convection strengthens due to the reduction of the vertical stratification associated with the freshening/weakening of the Iceland-Scotland overflow. Our study provides a scenario that the change in

the Labrador Sea open-ocean deep convection strength is not a cause of the AMOC change. On the contrary, the AMOC weakening is associated with an opposite change (i.e., strengthening) in the Labrador Sea open-ocean deep convection through the freshening along the NAC pathway in the eastern subpolar North Atlantic and the downstream Iceland-Scotland overflow pathways. The surface buoyancy loss is not the only factor affecting the Labrador Sea open-ocean deep convection strength. The density of ISOW and the associated downstream deep NEADW layer could also affect the vertical stratification and, thus, the deep convection strength in the central Labrador Sea. The pronounced freshening in the ISOW and the NEADW layer in the deep Labrador Sea is affected by the freshening in the upper eastern subpolar North Atlantic induced by the AMOC weakening. The weakening of the Iceland-Scotland overflow also contributes to the freshening in the ISOW and the NEADW layer in the deep Labrador Sea by reducing the advection of climatological mean state salty water. This deep freshening reduces the deep ocean density and the vertical stratification in the central

Labrador Sea, leading to enhanced open-ocean deep convection. This mechanism elucidates why open-ocean deep convection in the Labrador Sea strengthens whereas the AMOC declines.

Climate models often lack a good representation of the ISOW-associated NEADW layer in the deep Labrador Sea (Supplementary Fig. 9), and have too coarse resolutions to resolve the Labrador Sea boundary currents. A realistic representation of both the ISOW-associated NEADW layer and the narrow boundary current in the Labrador Sea is important to simulate the impact of the ISOW on the Labrador Sea deep convection under the external freshwater forcing. The freshwater anomalies advected by the much broader Labrador Sea boundary current in 1° coarse-resolution models would affect interior Labrador Sea unrealistically and these models are not suitable to study the impact of upstream freshwater anomalies on the downstream Labrador Sea[47,49]. In contrast, in 0.25° or higher resolution models, the freshwater anomalies are mainly confined within the substantially better-resolved narrow Labrador Sea boundary current with limited impact on interior Labrador Sea[47,49]. The model employed in this study has an improved representation of the ISOW-associated NEADW layer and the narrow boundary current in the Labrador Sea. Hence this model is capable to simulate the influence of the freshening/weakening of the Iceland-Scotland overflow on the strengthening of the Labrador Sea open-ocean deep convection. The 0.25° models are still not able to fully resolve eddies, and higher-resolution models would be needed in future studies to provide a more accurate representation of the eddy mixing process between the boundary and the interior Labrador Sea.

The intense Labrador Sea open-ocean deep convection that occurred in the early 1990s is indeed accompanied by the freshening in the upper eastern subpolar North Atlantic, the downstream ISOW, and the deep NEADW layer in the central Labrador Sea[33,50], in addition to the excessive surface heat loss. The observed strengthening of the Labrador Sea open-ocean deep convection from the late 1960s to the early 1990s[33,50] is also accompanied by the observed weakening/lightening of the ISOW over this period[51,52]. The influence of the ISOW density/strength and associated downstream NEADW layer density on the Labradors Sea open-ocean deep convection strength deserves more attention. A realistic representation of the ISOW-associated NEADW layer and the narrow boundary current in the Labrador Sea is necessary for future modeling studies on this topic.

## Methods
### The control and water-hosing experiments
The control and water hosing experiments in this study are conducted with an eddy-permitting coupled climate model (GFDL CM4[53]) with a hybrid vertical coordinate in the ocean component[53,54]. The external radiative forcings are fixed at 1990s conditions. The ocean model has about 0.25° horizontal resolution (eddy-permitting at mid latitudes while still too coarse at high latitudes) and the atmosphere model has about 100 km horizontal resolution. The ocean bathymetry around the Faroe Bank Channel is significantly deepened and widened to have an improved representation of the Iceland-Scotland overflow strength in the control simulation that is close to observations. The water hosing experiment has an idealized external freshwater flux of 0.05 Sv (1 Sv=$10^6$ m³ s⁻¹) uniformly distributed over the southern Nordic Sea (the blue box in Fig. 1d; 64°N ~ 69°N, 45°W ~ 20°E) for the entire 80-year duration of the experiment. The global mean salinity is not conserved under the external freshwater forcing without any compensation. Here the 0.05 Sv external freshwater flux is ~25% of the total observed climatological freshwater flux, ~0.2 Sv[55,56], (from all sources such as precipitation, river runoff, and Greenland ice sheet melt) that eventually enters the Nordic Sea and is exported into the North Atlantic through the southern boundary of the Nordic Sea, i.e. the Greenland-Scotland Ridge.

In some previous water hosing experiments, the external freshwater flux is released broadly over the entire subpolar North Atlantic and directly covers the interior Labrador Sea deep convection site, resulting in the weakening and shutdown of the Labrador Sea open-ocean deep convection[4,5]. However, observational and high-resolution modeling studies suggest that the freshwater flux enters the Labrador Sea mainly through its boundary current, and the freshwater flux is largely confined within the narrow Labrador Sea boundary current with limited access to the interior Labrador Sea deep convection site[47,49,57,58]. Hence it is less realistic to have freshwater flux released directly into the interior Labrador Sea deep convection site. Meanwhile, observational and high-resolution modeling studies suggest that the freshwater flux entering the Nordic Sea through the Fram Strait is advected southeastward into the interior southern Nordic Sea (northeast of Iceland) by the Jan Mayen Current (JMC) and the East Icelandic Current (EIC), and recirculates in the southern Nordic Sea[47,59]. Paleo sea ice proxy records from the North Icelandic shelf have been employed to reconstruct the anomalous Arctic freshwater flux/sea ice exported through the Fram Strait and accumulated in the southern Nodic Sea, and they reveal multidecadal periods of substantially enhanced freshwater flux/sea ice in the southern Nordic Sea during the past several hundreds of years[59]. Paleo records from the southern Norwegian Sea also suggest that anomalous Arctic freshwater outflow entered this region during the early stage of the last interglacial period[60]. Since there is no open-ocean deep convection in the southern Nordic Sea, the anomalous freshwater flux entering this region as discussed above will not directly affect any open-ocean deep convection. For the above reasons, we choose the southern Nordic Sea instead of the entire subpolar North Atlantic to apply the external freshwater flux. Our study aims to explore if it is possible to have an alternative relationship between the AMOC change and the Labrador Sea open-ocean deep convection change in response to an external freshwater forcing when the freshwater forcing is applied over the southern Nordic Sea where there is no open-ocean deep convection.

An ensemble of three 80-year water hosing members is conducted, and each ensemble member has a different initial condition taken from 20 years apart of the control simulation. The 'anomaly' in this paper refers to the ensemble-mean of the difference between the 80-year water hosing member and the corresponding 80-year control simulation segment with the same initial condition (water hosing – control). The ensemble mean is used throughout this study to reduce the effect of internal variability presented in each individual ensemble member and improve the signal-to-noise ratio, although some residual internal variability still appears in the three-member ensemble mean. For example, the residual decadal variability in the March MLD (Fig. 1c) is related to the residual decadal variability in the surface heat loss and associated surface/upper Labrador Sea density (Supplementary Fig. 8b). The residual decadal variability in the maximum AMOC across OSNAP West is in phase with the residual decadal variability in the Labrador Sea March MLD (Fig. 1b, c). This relationship might be overestimated due to modeling deficiencies in separating the boundary current from the deep convection region in the Labrador Sea.

In the control simulation, the maximum density-space AMOC exhibits a strength of 17.2 Sv across the OSNAP section averaged over the last 40 years of the ensemble, which is slightly stronger than that observed from the OSNAP[19]. The maximum density-space AMOC across the OSNAP West subsection averaged over the last 40 years of the ensemble is about 5.1 Sv, much larger than the observed value of around 2.6 Sv[19]. Supplementary Fig. 10 shows the comparison of modeled OSNAP AMOC streamfunctions in the control simulation averaged over the last 40 years of the ensemble with the OSNAP observations. The overestimation of the AMOC across the OSNAP West subsection in the control simulation is likely due to modeling deficiencies in separating the boundary current from the deep convection

region and representing the boundary density contrast in the Labrador Sea[18,20].

The modeled deep stratification in the central Labrador Sea, which is defined as the potential density ($\sigma_0$) difference between the ISOW-associated NEADW layer (2000–2500 m) and the core Labrador Sea Water (LSW) layer (1000–1500 m) above, is 0.063 kg m$^{-3}$ in the control simulation averaged over the last 40 years of the ensemble. This modeled long-term mean deep stratification value is very similar to that (0.061 kg m$^{-3}$) derived from the observed World Ocean Atlas 2018 (WOA18) data averaged over the past several decades (1955–2017).

## AMOC definition
The AMOC across the OSNAP section is calculated from the volume transport in density space so that it contains both the thermal wind contribution[4,61] (i.e., depth-space AMOC) and the contribution from the horizontal circulation across sloping isopycnals. We analyze the time series of both the maximum AMOC (defined as the maximum streamfunction in density space) and the fixed-level AMOC (defined as the streamfunction at a certain density level, e.g., at a density level where $\sigma_0 = 27.84$ kg m$^{-3}$) to understand the response of the AMOC to the external freshwater forcing.

## Water mass transformation
We analyze the long-term mean surface forced water mass transformation (WMT$_S$) and water mass transformation forced by interior mixing (WMT$_M$) similar to those shown in recent studies[23,24,62–64]. The results (Supplementary Fig. 3) are averaged over the last 40 years of the control and water hosing experiments for the entire region northeast of OSNAP East, the region between OSNAP East and the Greenland-Scotland Ridge (GSR), i.e. Iceland-Irminger Seas (IIS), and the region northeast of GSR. The surface forced water mass transformation (WMT$_S$) is calculated using the methods developed by Drake et al. 2024[64]. For these quasi-equilibrium long-term mean analyses, the WMT$_M$ is estimated as the difference between the density-space AMOC streamfunction across OSNAP East or GSR (or the difference between these two sections for the IIS region) and the corresponding WMT$_S$ (same approach used in Petit et al. 2023[24]).

## Density in the western and eastern parts of OSNAP East
We also analyze the temporal evolution of the density and the haline/thermal components of the deep ocean density in the western and eastern parts of OSNAP East respectively (Supplementary Fig. 5; Fig. 5). The western part of OSNAP East is defined as a narrow area along the western boundary, whereas the eastern part of OSNAP East encompasses a larger area (as indicated by green boxes in Fig. 1d). These areas were chosen according to locations of the AMOC inflow and outflow, and the results are not sensitive to the selection of areas. The relationship between the density-space AMOC and the west–east density contrast is much more complicated in reality. Here the diagnosed deep ocean west–east density contrast is interpreted as a simplified indicator[62,65] for the transient evolution of the density-space AMOC decline across OSNAP East at the relatively dense level, which is dominated by the reduced contribution from the horizontal circulation across sloping isopycnals.

## Freshwater fraction
The salt-based freshwater fraction (FWF; unitless) is defined as

$$FWF_{salt} = \frac{S_r - S}{S_r} \qquad (1)$$

Here $S_r$ is the reference salinity of 35 psu, and $S$ is the salinity in each grid cell. Hence the salt-based freshwater fraction anomaly

($FWF'_{salt}$) is calculated as

$$FWF'_{salt} = -\frac{S'}{S_r} \qquad (2)$$

Where $S'$ is the salinity anomaly (water hosing–control) in each grid cell.

A passive dye tracer flux is added with the same amplitude/spatial distribution as the external freshwater flux in the water hosing experiments to trace the propagation of the externally forced freshwater anomaly. Hence the dye-based freshwater fraction anomaly ($FWF'_{dye}$) is represented by the passive dye tracer concentration $C_{tr}$ (unitless), i.e.,

$$FWF'_{dye} = C_{tr} \qquad (3)$$

The salt-based FWF anomaly $FWF'_{salt}$ reflects the total FWF anomaly. Meanwhile, the dye-based FWF anomaly $FWF'_{dye}$ only reflects the direct contribution from the external freshwater forcing, i.e., the advection and diffusion of the externally forced freshwater anomaly. Hence the difference between the salt-based FWF anomaly and the dye-based FWF anomaly ($FWF'_{salt} - FWF'_{dye}$) reflects the additional FWF anomaly induced by changes in ocean circulation and mixing acting on the climatological FWF and the coupled feedback through the advection/diffusion of the additional FWF anomaly.

## Data availability
The World Ocean Atlas 2018 (WOA18) data were downloaded from the NOAA National Centers for Environmental Information (formerly the National Oceanographic Data) https://www.ncei.noaa.gov/products/world-ocean-atlas/. The Data from the OSNAP (Overturning in the Subpolar North Atlantic Program) array were downloaded from https://www.o-snap.org/. OSNAP data were collected and made freely available by the OSNAP project and all the national programs that contribute to it (www.o-snap.org). The CMIP6 (Coupled Model Intercomparison Project Phase 6) model data were downloaded from https://aims2.llnl.gov/search/cmip6/. The key data used in this study are available in the Zenodo dataset at https://doi.org/10.5281/zenodo.13370392[66].

## Code availability
The source code of the Geophysical Fluid Dynamics Laboratory (GFDL) coupled climate model version 4 (CM4) is publicly available at https://doi.org/10.5281/zenodo.3339397. The surface forced water mass transformation (WMT$_S$) is calculated using the source code developed by Drake et al. 2024[64] at https://github.com/hdrake/xwmt.

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

## Acknowledgements
Bob Hallberg, Matt Harrison, and Brandon Reichl are acknowledged for their helpful comments and suggestions on this work. X.W. is supported by Princeton University Graduate School and CIMES Task II funds for graduate research under award NA18OAR4320123 from the National Oceanic and Atmospheric Administration, U.S. Department of Commerce. R.Z. is supported by GFDL base funding. We acknowledge GFDL resources made available for this research. We acknowledge the World Climate Research Programme, which, through its Working Group on Coupled Modeling, coordinated and promoted CMIP6. We thank the climate modeling groups for producing and making available their model output, the Earth System Grid Federation (ESGF) for archiving the data and providing access, and the multiple funding agencies who support CMIP6 and ESGF.

## Author contributions
R.Z. conceived the study and designed/conducted the numerical experiments. X.W. conducted the analyses of the numerical experiments. The authors discussed the results/interpretations and wrote the paper together.

## Competing interests
The authors declare no competing interests.
