## [Transparent Peer Review file · Nature Communications]

Weakening of the AMOC and Strengthening of Labrador Sea Deep Convection in Response to External Freshwater Forcing

Corresponding Author: Ms Xinyue Wei

Version 0:

Reviewer comments:

Reviewer #1

(Remarks to the Author)

Please find comments in the attachment.

Reviewer #2

(Remarks to the Author)

Thank you for the manuscript "The weakening of the AMOC and associated strengthening of the Labrador Sea open-ocean deep convection". I think there are some interesting results here, in particular that a freshening of the Nordic seas can lead to a strengthening of the Labrador sea convection. I'm not convinced though that these results would be of interest to a wider audience. There are also some major comments below.

Major

There's quite a lot of reliance on west-east density changes, however there are a few problems with the way the authors are dealing with these. Theory shows that the AMOC strength in depth space is related to the pressure difference from one boundary to the other. So the authors should be depth-integrating the densities at the boundaries rather than averaging the densities over a wide range of longitudes. Also the theory is for AMOC in depth space, rather than density space, so it doesn't capture all the AMOC signal. A better way to link the AMOC and properties in density space would be to calculate the water mass transformation indicated by the surface fluxes.

The structure seems to be "here's a few pieces of analysis we did on these experiments" rather than being a narrative of the results and feels rather disjointed. What is the key story you're telling and what are the key pieces of evidence you need to present to tell it? It's not clear what the relevance of the section about "thermal wind and horizontal circulation" is for the rest of the paper.

Minor

L12-15 This sentence is difficult to follow, particularly before these concepts have been explained.

L37-44 The relationship between deep convection and the AMOC is not simple (ie convection isn't a sinking of water, but mixes densities which drives density gradients which allow sinking along the boundary). This should be discussed (along with literature such as ref 1,2,3)

L62-72 What about waters from the Denmark Strait overflow which also make it into the Labrador Sea?

L104-107 Deep convection increases along the northern boundary of the Labrador and Irminger seas. Please discuss this. Why does this happen?

L144 There does seem to be a relationship between the decadal variability of the AMOC across OSNAP west and the Lab MLD.

L123-125 It doesn't look like the density change in the eastern deep ocean is negligible from Fig 3f. It certainly looks large below 2000m

L152 "which on average is around 2000m in the Labrador Sea"

L162-163 What is the 'cyclonic abyssal circulation'? Where is it?

L233 Looks more like 60 years to me

L250-251 Is the difference because of the mean state (and hence difference thermal/saline expansion coeff) or because of

different changes in the T/S ratio?

L286-289 I can't see this in Suppl Fig 5, though can in Fig 6.

L343 "pathway by mixing with the warm and saline water..."

L360 What do you mean by "downward propagation"? Is this from mixing?

L452 Why this hosing region?

L473-477 I agree that the representation of the boundary currents in climate models is a problem, however I disagree with this statement. Actually there is a wide range of strengths of OSNAP west in CMIP6 models (see ref 4), and it isn't obviously true that CMIP6 models generally overestimate it. It also isn't obvious true that this is related to resolution. For instance there are two resolutions of the HadGEM3 GC3.1 model in that reference, but it is the higher resolution model which has the larger Labrador sea overturning.

L520 I think you need to say where it will be made available.

References

1 [https://doi.org/10.1175/1520-0485\(2004\)034<1197:BCAWTI>2.0.CO;2](https://doi.org/10.1175/1520-0485(2004)034<1197:BCAWTI>2.0.CO;2)

2 https://journals.ametsoc.org/view/journals/phoc/31/3/1520-0485_2001_031_0810_wddwsa_2.0.co_2.xml

3 <http://dx.doi.org/10.1175/jpo2932.1>

4 <https://link.springer.com/article/10.1007/s00382-022-06448-1>

Reviewer #3

(Remarks to the Author)

See attached document.

Version 1:

Reviewer comments:

Reviewer #1

(Remarks to the Author)

One minor suggestion:

The rationale of releasing freshwater in the southern Nordic Sea has been elaborated in the Methods section. Suggest including a transition sentence summarizing previous freshwater release experiments in the Labrador Sea before Line 466, and then describe their limitations.

Reviewer #2

(Remarks to the Author)

Thank you for the manuscript "The weakening of the AMOC and associated strengthening of the Labrador Sea open-ocean deep convection". The authors have done a good job of addressing comments in their responses, however this doesn't really come across in the revised manuscript. Hence, there are a few major points that I will raise again – this is an issue about presentation, rather than the science, so I don't expect that they will take long to address.

Major

I'm still not at all clear why the authors chose this region to apply to hosing. I see that some discussion has been added, but it still doesn't seem to say anywhere 'we chose this region because of ...'. Is it supposed to represent some real process or targeted at something in particular, or just be a sensitivity study?

The authors responded with a clear answer when I asked about the main message of their paper, however there are still points in the manuscript where I got confused about what they were doing. I think this could be helped with a little reorganization and trying to bridge between topics. I would start with presenting the main results of AMOC weakening and MLD strengthening and using this to pose the main question of the paper. Then discuss the AMOC changes in more detail (including moving 'The transient AMOC decline section' earlier), followed, as they have done, by the tracer results and role of ISOW. I don't think this is a major change to what they have done and means that we're not jumping between different topics. The page or so from L138 to 176 is very confusing. There is a lot of jumping around topics. Lines 138-156 seem very much related to the transient AMOC discussion, so I suggest that they are removed or amalgamated (see also next point). The authors also need to point the author to why they are doing something when they change to a different topic. This is done in some of the later sections but not in the first two subsections of the results.

I still have concerns around the analysis of density gradients in lines 138-156. In their previous response to my questions about this, the authors said that they were not relying on the usual relationship between depth space AMOC and boundary density/pressures, but instead on a relationship between density space AMOC and density gradients from another paper. Firstly, this isn't clear in the revised text – instead the authors refer generally to the relationship between the AMOC and densities, referring to a number of papers, rather than making it clear that they are not relying on the usual arguments. Secondly, the papers they refer to (Zhang and Thomas 2021 and Arthun et al 2023) do not demonstrate a definitive relationship between the AMOC in density space and density gradients. I certainly couldn't see anything to suggest that the AMOC at a specific deep density levels was related to density changes in specific regions. Although the AMOC change in

density space is quite likely related to density changes, it isn't clear which regions are important for a given 'relatively denser' layer. We know that depth and density space AMOC are different at these latitudes, so they can't both have a simple relationship to zonal density gradients. To be honest I'd delete this paragraph and bring the 'Transient AMOC decline' subsection forwards.

Minor

L15 and throughout. The author's keep using the term 'the relatively denser level' and similar throughout the paper. I'm not a grammar expert, but this doesn't sound right to me (I'd say 'denser level' or 'relatively dense level'). There's also the issue that it's rather vague – it's difficult to tell whether this is a given level that is referred to throughout (in which case it might be best to define it as a symbol like ρ_0), or just that it's 'relatively' dense (in comparison to what?).

L107 How does the AMOC weakening compare to the variability in the control? The same goes for the MLD strengthening – is it significant compared to variability, or is it just that it doesn't weaken?

L109 What is meant here by significant? Is there a statistical significance test?

L131,134 'due' and 'caused' – Given the indirect relationship between the AMOC and the LS convection, we wouldn't expect one to cause the other. Maybe you want to say they are not related or similar?

L160 'Results are also consistent with observational analyses that suggest that the inferred historical weak AMOC phase is associated with...'

L162-176. You need to introduce what the WMT is first and what you're wanting to do with the analysis. Also reference to the methods for calculating it.

L215 upper Nordic Sea

L219-228 This is certainly not obvious from what is shown. Fig 6 doesn't show the Nordic seas, so we can't see how much dye there is there going over the overflows. Also could there not be sinking/mixing from the upper to lower waters along the boundary of the Irminger and Labrador seas?

L291-294 I don't think this is necessarily true. There is also a signal from dye originating from the Denmark Straits, but also it's not obvious that it couldn't partly come from the upper LS as well (see previous comment).

L340-344 Is this shown? Please add reference to figure or 'not shown' if you didn't include it.

L373-385 These two paragraphs seem to be very much stuck on the end. I'd suggest you find more relevant places to put each one.

L373-379 It would be interesting to include density anomalies of upper and lower densities in Supp Fig 7. Is it true that the variability is coming from the upper layer and trend from the lower layer or is it more complex?

L380-385 I think the last sentence could be expanded on a bit, including reference to figures. Is this freshening seen in the tracer for example? Eg is it directly from the forcing?

L427 "... in the deep Labrador Sea and boundary currents, both of which are important to represent this process" Or similar

L431-433 However 0.25 degree models are not able to simulate eddies which mix between the boundary and interior LS. So this is a caveat for whether this model is properly representing this process.

L483-4 There is quite a lot of repetition.

L506 Normally I think of stratification as being from the surface layer. You should note that this is deep stratification.

Reviewer #3

(Remarks to the Author)

I would like to thank the authors for their replies to all the reviewers. The clarifications, additional analyses and results helped to strengthen the main messages in the revised manuscript. I would also like to thank the authors for their analysis on the CMIP6 data, I know that it can be quite a challenge in analysing the different CMIP6 models. These CMIP6 results prompt some further questions, but they are beyond the scope of this manuscript.

I'm happy to recommend the manuscript for publication.

Version 2:

Reviewer comments:

Reviewer #2

(Remarks to the Author)

The authors have addressed my comments so I'm happy to recommend accepting the manuscript

Response to Review Comments

Response to Reviewer #1

The study investigates the responses of AMOC and Labrador Sea convection to external freshwater forcing applied in the southern Nordic Sea in an eddy-permitting coupled climate model. They found a decline of the AMOC associated with reduced zonal density contrast across the subpolar North Atlantic, which was further attributed to both direct impact from freshwater anomalies and indirect contributions from circulation/mixing changes. In contrast to a weakening AMOC, they found a strengthening convection in the Labrador Sea. The enhanced convection was further explained by decreased density (enhanced freshening) in the overflow water layer that reduced the vertical stratification. These results suggest that Labrador Sea convection is not the cause of the AMOC decline.

I find the study very interesting and unique. The finding that the AMOC and Labrador Sea convection have opposite trends under external freshwater forcing is novel, and the conclusion that weakened AMOC leads to enhanced convection by reducing vertical stratification is noteworthy. The manuscript is overall well-written, and the presentations of the results are clear. That being said, I do have several concerns in terms of the freshwater release experiment design and the model results validations, which I believe could be improved by further discussions and analyses.

We thank the Reviewer for the very positive and constructive comments/suggestions. We have revised the manuscript accordingly and below is our detailed response to the comments and suggestions.

Major comments

(1). The rationale of the freshwater release location needs to be elaborated. Results of the freshwater hosing experiments may depend on where the external freshwater flux is injected. Previous studies uniformly inject freshwater over the Labrador Sea and thus find a causal relationship between convection and AMOC. The current study, on the other hand, releases freshwater in the southern Nordic Sea and finds opposite trends of AMOC and Labrador Sea convection. Which freshwater release site is more realistic? Is there observational evidence that supports enhanced freshwater flux over southern Nordic Sea?

Observational and high-resolution modelling studies suggest that the freshwater flux enters the Labrador Sea mainly through its boundary current, and the freshwater flux is largely confined within the narrow Labrador Sea boundary current with limited access to the interior Labrador Sea deep convection site (e.g., Schmidt and Send, 2007; Marsh et al., 2010; Florindo-Lopez et al., 2020; Kim et al., 2021). Meanwhile, observational and high-resolution modelling studies suggest that the freshwater flux entering the Nordic Sea through the Fram Strait is advected southeastward into the interior southern Nordic Sea (northeast of Iceland) by the Jan Mayer Current (JMC) and

the East Icelandic Current (EIC), and recirculates in the southern Nordic Sea (Miles et al., 2020; Kim et al., 2021). Hence comparing these two freshwater release sites used in modelling studies, it is less realistic to have freshwater flux released directly into the interior Labrador Sea deep convection site. Paleo sea ice proxy records from the North Icelandic shelf have been employed to reconstruct the anomalous Arctic freshwater flux/sea ice exported through the Fram Strait and accumulated in the southern Nordic Sea, and they reveal multidecadal periods of substantially enhanced freshwater flux/sea ice in the southern Nordic Sea during the past several hundreds of years (Miles et al. 2020). We have added more related elaboration in the revised Methods section.

We have also clarified in the revised Methods section that “Our study aims to explore if it is possible to have an alternative relationship between the AMOC change and the Labrador Sea open-ocean deep convection change in response to an external freshwater forcing, when the freshwater forcing is applied over the southern Nordic Sea where there is no open-ocean deep convection.”

(2). The Labrador Sea stratification or mixed layer depth in control simulation needs to be validated with observations. The conclusion that overflow layer density impacts convection by modifying vertical stratification is valid when convection frequently reaches the deep overflow layer. However, observed convection in the Labrador Sea is ~1000-1500m (except for that in the early 1990s), which is above the overflow layer and is less influenced by the overflow density change. If the model overestimates convection depth, the impact from overflow layer density could very likely be exaggerated.

Thanks for the suggestion. The observations (van Aken et al., 2011, Figure 3c,d) suggest that in the central Labrador Sea over the past six decades (1950-2010), the core Labrador Sea Water (LSW) layer (around 1000-1500m) is often characterized with high oxygen due to convective ventilation and is above the Iceland-Scotland overflow water (ISOW)-induced Northeast Atlantic Deep Water (NEADW) layer (around 2000-2500m) characterized with low oxygen. We have validated with observations - World Ocean Atlas 2018 (WOA18) data for the long-term mean vertical stratification in the central Labrador Sea, which is defined as the potential density (σ_0) difference between the ISOW-associated NEADW layer (2000-2500m) and the core LSW layer (1000-1500m) above. The modeled long-term mean vertical stratification in the central Labrador Sea is 0.063 kg/m^3 in the control simulation, which is very similar to that (0.061 kg/m^3) derived from the observed World Ocean Atlas 2018 (WOA18) data averaged over the past several decades (1955-2017). We have discussed this validation in the revised Methods section. We agree that the strength of the ISOW-induced Labrador Sea deep convection increase could be sensitive to this background vertical stratification (density difference) in the central Labrador Sea. For example, if the density of the ISOW-associated NEADW layer is reduced and becomes less than the density of the core LSW layer, the vertical stratification in the central Labrador Sea will become completely unstable and trigger a much deeper convection.

(3). Further discussion on the relationship between Labrador Sea convection and the AMOC is suggested. As stated in the manuscript, the OSNAP observations have revealed that overturning in the eastern basin dominates the subpolar AMOC. In the current study, it is further shown that AMOC decline in response to freshwater forcing is determined by circulation and property changes in the eastern subpolar basin. How to reconcile these results with previous studies that suggest the AMOC variability determined by Labrador Sea convection? Is it a matter of time scales? I think the relationship between convection and the AMOC worth in-depth discussion.

We have added the following in-depth discussion on the relationship between Labrador Sea deep convection and the AMOC in the revised Introduction section as suggested:

“Previous studies have shown that the net sinking induced by the Labrador Sea open-ocean deep convection is negligible, the contribution to the AMOC from the Labrador Sea is mainly from boundary sinking, and the impact of the open-ocean deep convection on the AMOC has to be indirect through eddy mixing with boundary properties (Send and Marshall, 1995; Boning et al., 1996; Spall and Pickart, 2001; Spall, 2004; Straneo, 2006; Pickart and Spall, 2007; Holte and Straneo, 2017; Georgiou et al., 2019). Hence the AMOC (associated with the net sinking) and open-ocean deep convection (with little direct contribution to the net sinking) involve different physical processes.”

“A recent study (Lai et al., 2022) shows that the relative role of the Labrador Sea deep convection in multidecadal AMOC variability is model dependent: in the model with unrealistically strong/wide mean state Labrador Sea deep convection, multidecadal AMOC variability is dominated by multidecadal variability in the Labrador Sea deep convection; whereas in the model with a relatively weak/narrow mean state Labrador Sea deep convection, multidecadal AMOC variability is dominated by multidecadal Arctic salinity variability. The model with unrealistically strong/wide mean state Labrador Sea deep convection has a higher-than-observed density along the Labrador Sea boundary current and overestimates the mean state AMOC across OSNAP West (Jackson et al., 2020; Jackson and Petit, 2023; Petit et al., 2023). The dominant role of the Labrador Sea deep convection in multidecadal AMOC variability might also be overestimated in models with overestimated mean state AMOC across OSNAP West due to modelling deficiencies in separating the boundary current from unrealistically strong/wide deep convection in the Labrador Sea.”

Minor comments

(1). Please indicate in the title that the changes are in response to external freshwater flux.

Done.

(2). Lines 146-158: It is a bit difficult to tell horizontal circulation change just by looking at Fig.2. Reporting magnitudes of AMOC change in density and depth space would help.

We have added the magnitudes of the AMOC change in density (5.7Sv) and depth (0.8Sv) space here as suggested.

(3). Line 163: I would suggest using abyssal circulation instead of recirculation.
Done.

(4). Line 195: I am confused by how freshwater anomalies get into the deep ocean. I thought the fresh anomalies would mostly stay in the near-surface layer, and most of the mixing and entrainment occurred in the bottom layer. In other words, the freshwater anomalies cannot be directly entrained into the deep layer.

The eastern Nordic Sea is a region with surface dense water formation along the Atlantic inflow pathway. Some surface freshwater anomalies get into the relatively deeper layer in the eastern Nordic Sea through surface dense water formation there, then recirculate in the Nordic Sea, and move out of the Nordic Sea with the overflows. The descending Nordic Sea overflows carry these freshwater anomalies at the relatively deeper layer of the Greenland-Scotland Ridge directly into the deep subpolar North Atlantic along the overflow pathways (i.e. through advection). We have added more clarification here in the revised text.

(5). Fig.1: There is an interesting decadal variability of AMOC at OSNAP-West, which seems to be associated with Labrador Sea MLD decadal variability. Is this relationship present in the control simulation?

The correlation between the decadal variations of the maximum AMOC across OSNAP West and the Labrador Sea March MLD in the control simulation is low (0.36) and not statistically significant. Meanwhile, the correlation between the residual decadal variations of these two variables in the ensemble mean of the control simulation segments is 0.7 and statistically significant. The ensemble mean reduces the internal variability that is independent between the two variables and thus the correlation between the residual decadal variability of the two variables is much higher.

We have added the following discussion of the relationship between the residual decadal variability of the two variables in the revised text (right after the discussion of the residual decadal variability in the Labrador Sea MLD in the original text Lines 379-381): “The residual decadal variability in the maximum AMOC across OSNAP West is in phase with the residual decadal variability in the Labrador Sea March MLD (Fig. 1b, c). This relationship might be overestimated due to modelling deficiencies in separating the boundary current from the deep convection region in the Labrador Sea.”

reference

1. Schmidt, S., & Send, U. (2007). Origin and composition of seasonal Labrador Sea freshwater. *Journal of Physical Oceanography*, 37(6), 1445-1454.

2. Marsh, R., Desbruyères, D., Bamber, J. L., De Cuevas, B. A., Coward, A. C., & Aksenov, Y. (2010). Short-term impacts of enhanced Greenland freshwater fluxes in an eddy-permitting ocean model. *Ocean Science*, *6*(3), 749-760.
3. Florindo-López, C., Bacon, S., Aksenov, Y., Chafik, L., Colbourne, E., & Holliday, N. P. (2020). Arctic Ocean and Hudson Bay freshwater exports: New estimates from seven decades of hydrographic surveys on the Labrador Shelf. *Journal of Climate*, *33*(20), 8849-8868.
4. Kim, W. M., Yeager, S., & Danabasoglu, G. (2021). Revisiting the causal connection between the great salinity anomaly of the 1970s and the shutdown of Labrador Sea deep convection. *Journal of Climate*, *34*(2), 675-696.
5. Miles, M. W., Andresen, C. S., & Dylmer, C. V. (2020). Evidence for extreme export of Arctic sea ice leading the abrupt onset of the Little Ice Age. *Science advances*, *6*(38), eaba4320.
6. Pickart, R. S., & Spall, M. A. (2007). Impact of Labrador Sea convection on the North Atlantic meridional overturning circulation. *Journal of Physical Oceanography*, *37*(9), 2207-2227.
7. Holte, J., & Straneo, F. (2017). Seasonal overturning of the Labrador Sea as observed by Argo floats. *Journal of Physical Oceanography*, *47*(10), 2531-2543.
8. Send, U., & Marshall, J. (1995). Integral effects of deep convection. *Journal of physical oceanography*, *25*(5), 855-872.
9. Böning, C. W., Bryan, F. O., Holland, W. R., & Döscher, R. (1996). Deep-water formation and meridional overturning in a high-resolution model of the North Atlantic. *Journal of Physical Oceanography*, *26*(7), 1142-1164.
10. Spall, M. A., & Pickart, R. S. (2001). Where does dense water sink? A subpolar gyre example. *Journal of Physical Oceanography*, *31*(3), 810-826.
11. Spall, M. A. (2004). Boundary currents and watermass transformation in marginal seas. *Journal of physical oceanography*, *34*(5), 1197-1213.
12. Straneo, F. (2006). On the connection between dense water formation, overturning, and poleward heat transport in a convective basin. *Journal of Physical Oceanography*, *36*(9), 1822-1840.
13. Georgiou, S., van der Boog, C. G., Brüggemann, N., Ypma, S. L., Pietrzak, J. D., & Katsman, C. A. (2019). On the interplay between downwelling, deep convection and mesoscale eddies in the Labrador Sea. *Ocean Modelling*, *135*, 56-70.
14. Lai, W. K. M., Robson, J. I., Wilcox, L. J., & Dunstone, N. (2022). Mechanisms of internal Atlantic multidecadal variability in HadGEM3-GC3. 1 at two different resolutions. *Journal of Climate*, *35*(4), 1365-1383.
15. Jackson, L. C., Roberts, M. J., Hewitt, H. T., Iovino, D., Koenig, T., Meccia, V. L., ... & Wood, R. A. (2020). Impact of ocean resolution and mean state on the rate of AMOC weakening. *Climate Dynamics*, *55*(7), 1711-1732.
16. Jackson, L. C., & Petit, T. (2023). North Atlantic overturning and water mass transformation in CMIP6 models. *Climate Dynamics*, *60*(9), 2871-2891.

17. Petit, T., Robson, J., Ferreira, D., & Jackson, L. C. (2023). Understanding the Sensitivity of the North Atlantic Subpolar Overturning in Different Resolution Versions of HadGEM3 - GC3.1. *Journal of Geophysical Research: Oceans*, 128(10), e2023JC019672.

Response to Reviewer #2

Thank you for the manuscript “The weakening of the AMOC and associated strengthening of the Labrador Sea open-ocean deep convection”. I think there are some interesting results here, in particular that a freshening of the Nordic Seas can lead to a strengthening of the Labrador sea convection. I am not convinced though that these results would be of interests to a wider audience. There are also some major comments below.

We thank the Reviewer for the very constructive comments/suggestions. We have revised the manuscript accordingly to address the Reviewer’s comments/suggestions. In particular, we have added the analysis of the surface and interior mixing forced water mass transformation and the related figure in the revision.

As far as we know, there are rarely any modeling studies on the impact of the freshening/weakening of the Iceland-Scotland overflow water (ISOW) on the strengthening of the Labrador Sea open-ocean deep convection. Many climate models have difficulties in simulating the ISOW-associated NEADW layer in the Labrador Sea in their control simulations, and thus are not able to simulate this phenomenon. The important impact of the ISOW density/strength and associated downstream NEADW layer on the Labradors Sea open-ocean deep convection strength deserves wider attention. The novel and unique results of this study will call for wider attention to this area.

Major

There’s quite a lot of reliance on west-east density changes, however there are a few problems with the way the authors are dealing with these. Theory shows that the AMOC strength in depth space is related to the pressure difference from one boundary to the other. So the authors should be depth-integrating the densities at the boundaries rather than averaging the densities over a wide range of longitudes. Also the theory is for AMOC in depth space, rather than density space, so it doesn’t capture all the AMOC signal. A better way to link the AMOC and properties in density space would be to calculate the water mass transformation indicated by the surface fluxes.

Thanks for the suggestion. This study is focused on the density-space AMOC rather than the depth-space AMOC across the subpolar North Atlantic, because at such high latitudes the depth-space AMOC is not suitable to represent the full AMOC signal revealed by the density-space AMOC as

suggested by previous studies (e.g. Zhang, 2010; Li et al., 2019) and also by the Reviewer. Hence the discussion of the west-east density changes in this study is not aimed for the depth-space AMOC, but for the transient evolution of the density-space AMOC decline. In particular, it is aimed to understand the contribution of the subpolar horizontal circulation across sloping isopycnals, which is affected by the west-east density contrast of the subpolar horizontal circulation (Zhang and Thomas, 2021), to the transient evolution of the density-space AMOC decline. Although the outflow of the subpolar horizontal circulation is concentrated in the western boundary, the inflow of the subpolar horizontal circulation extends into the interior ocean in the eastern basin. Hence, we use a relatively wider range of longitudes for the eastern basin density box to cover the inflow of the subpolar horizontal circulation. Similarly, a recent paper (Arthun et al., 2023) uses of the west-east density contrast as a simplified indicator for the horizontal circulation contribution to the transient evolution of the density-space AMOC changes, and also uses a relatively wider range of longitudes to calculate the west-east density contrast. We have made more clarifications of the above points in the revision. To focus on the key message and save more space for the revision, we have deleted the discussion of the upper ocean west-east density contrast, and only kept the discussion of the deep ocean west–east density contrast as a simplified indicator for the transient evolution of the density-space AMOC decline at the relatively denser level across OSNAP East.

In addition to the above transient time-dependent analyses, following the Reviewer’s suggestion, we have included analyses of the long-term mean surface forced water mass transformation (WMTs) similar to those shown in previous studies (e.g., Jackson and Petit, 2023; Petit et al. 2023; Arthun et al., 2023; Tesdal et al., 2023; Drake et al., 2024) in the new Supplementary Fig. 3 of the revision. The WMTs are averaged over the last 40 years of the control and water hosing experiments for the entire region northeast of OSNAP East, the region between OSNAP East and the Greenland-Scotland Ridge (GSR), i.e. Iceland-Irminger Seas (IIS), and the region northeast of GSR. We have also plotted the long-term mean water mass transformation forced by interior mixing (WMT_M) for the above three regions in the same figure. For these quasi-equilibrium long-term mean analyses, the WMT_M is estimated as the difference between the density-space AMOC streamfunction across OSNAP East or GSR (or the difference between these two sections for the IIS region) and the corresponding WMTs (same approach used in Petit et al., 2023).

The new Supplementary Fig. 3 shows that the long-term mean density-space AMOC decline across GSR (panel c) is affected by both the long-term mean WMTs (surface forced) and WMT_M (interior mixing forced) anomalies northeast of GSR that often counter each other at various density levels (panel i). In the IIS region, the long-term mean total WMT anomaly (i.e. AMOC anomaly across OSNAP East – AMOC anomaly across GSR) is mainly dominated by the WMT_M (interior mixing forced) anomaly, with little change in the WMTs (surface forced) anomaly (panel b, h). The long-term mean density-space AMOC decline across OSNAP East (panel a) is mainly balanced by the long-term mean WMTs (surface forced) decline occurring in large part remotely northeast of GSR at the relatively denser level (27.8 kg/m^3) (panel a, g), because the long-term mean WMT_M (interior

mixing forced) anomalies for the IIS region and the region northeast of GSR are almost cancelled at this density level. The above long-term mean balance does not necessarily apply to the transient evolution: at the relatively denser level, the WMTs decline northeast of GSR reaches equilibrium much earlier around 20 years, whereas the AMOC decline across OSNAP East in the corresponding deep ocean has a much longer decline timescale of more than 40 years (Fig. 7c, shown in the “The transient AMOC decline” section). We have added related discussion of the above surface and interior mixing forced water mass transformation results in the revision.

The structure seems to be “here’s a few pieces of analysis we did on these experiments” rather than being a narrative of the results and feels rather disjointed. What is the key story you’re telling and what are the key pieces of evidence you need to present to tell it? It’s not clear what the relevance of the section about “thermal wind and horizontal circulation” is for the rest of the paper.

Thanks for the questions. The key story of this study is the opposite changes between the AMOC and the Labrador Sea deep convection (including the impact of the ISOW on the Labrador Sea deep convection) under the external freshwater forcing. The key pieces of evidence include the time-mean and transient evolution of the AMOC, Labrador Sea deep convection, seawater properties, salt- and dye- based FWF anomalies as well as analyses designed to understand distinct mechanisms/processes involved in the opposite changes between the AMOC and the Labrador Sea deep convection, and the impact of the ISOW on the Labrador Sea deep convection. To focus on the key story as the Reviewer suggested and save more space for the revision, we have deleted the discussions of the upper ocean west-east density contrast and significantly shortened the section about “thermal wind and horizontal circulation” in the revision. We only kept the horizontal circulation contribution part of this section to explain why there is a substantial density-space AMOC decline but little depth-space AMOC decline across OSNAP East, and the distinct process of the horizontal circulation contribution to the density-space AMOC decline at the relatively denser level.

Minor

L12-15 This sentence is difficult to follow, particularly before these concepts have been explained.

This sentence has been revised and simplified by removing the concepts and making it easier to understand.

L37-44 The relationship between deep convection and the AMOC is not simple (i.e. convection isn’t a sinking of water, but mixes densities which drives density gradients which allow sinking along the boundary). This should be discussed (along with literature such as ref 1,2,3)

We have added here more discussion on the relationship between the AMOC and the deep convection as suggested and cited more related references (including ref 1,2,3 suggested by the Reviewer).

L62-72 What about waters from the Denmark Strait overflow which also make it into the Labrador Sea?

We have moved the introduction of the Denmark Strait overflow and its entering into the Labrador Sea from Lines 365-369 in the original manuscript (Results Section) to here in the Introduction section as suggested.

L104-107 Deep convection increases along the northern boundary of the Labrador and Irminger seas. Please discuss this. Why does this happen?

We think you mean the March MLD decreases along the northern boundary of the Labrador and Irminger Seas. We discussed the March MLD response at the northern boundary of the Labrador Sea in the original manuscript at Lines 382-387: “The March MLD is reduced over the northern boundary of the Labrador Sea and the southeastern boundary of the Nordic Sea (Fig. 1e). The climatological winter convection is often much shallower (Fig. 1d) in these regions mainly over continental slopes. The surface/upper ocean density decline induced by the freshening along the boundary currents dominates the reduction of March MLD over these continental slope regions.” We have added the northern boundary of the Irminger Sea in the above discussion in the revision.

L144 There does seem to be a relationship between the decadal variability of the AMOC across OSNAP west and the Lab MLD.

We agree with the Reviewer’s comment. The residual decadal variability in the Labrador Sea March MLD was discussed in the original manuscript Lines (378-381). We have added the following discussion right after the original Lines 378-381: “The residual decadal variability in the maximum AMOC across OSNAP West is in phase with the residual decadal variability in the Labrador Sea March MLD (Fig. 1b, c). This relationship might be overestimated due to modelling deficiencies in separating the boundary current from the deep convection region in the Labrador Sea.”

The correlation between the decadal variations of the maximum AMOC across OSNAP West and the Labrador Sea March MLD in the control simulation is low (0.36) and not statistically significant. Meanwhile, the correlation between the residual decadal variations of these two variables in the ensemble mean of the control simulation segments is 0.7 and statistically significant. The ensemble mean reduces the internal variability that is independent between the two variables and thus the correlation between the residual decadal variability of the two variables is much higher.

L123-125 It doesn’t look like the density change in the eastern deep ocean is negligible from Fig 3f. It certainly looks large below 2000m

We have clarified this by replacing the words “in the eastern side” with “near the eastern

boundary”, which refers to a region that is further east and away from the eastern flanks of the Reykjanes Ridge. We have also changed the word “negligible” to “relatively smaller”.

L152 “which on average is around 2000m in the Labrador Sea”

We have added the suggested clarification.

L162-163 What is the ‘cyclonic abyssal circulation’? Where is it?

We have added the new Supplementary Fig. 4 in the revision to show the climatological mean velocity across the OSNAP section. Across OSNAP East, the cyclonic abyssal circulation includes the southward western boundary current and the northward current near the eastern boundary at the same depth in the deep ocean. We have also added clarifications in the revised text.

L233 Looks more like 60 years to me

This sentence has been modified to “... decline rapidly within the first 40 years, then decrease slowly and reach equilibrium around year 60”.

L250-251 Is the difference because of the mean state (and hence difference thermal/saline expansion coeff) or because of different changes in the T/S ratio?

The difference is because of the mean state and hence the different thermal/saline expansion coeff. As explained in our response to the major comments, we have removed this discussion here to focus on the key message and save space for the revision.

L286-289 I can’t see this in Suppl Fig 5, though can in Fig 6.

We have changed the reference here to Fig. 6 instead of Suppl Fig.5 in the revision.

L343 “pathway by mixing with the warm and saline water...”

This sentence has been changed as suggested.

L360 What do you mean by “downward propagation”? Is this from mixing?

We mean both advection and mixing. We have replaced “propagation” here with “advection/diffusion” in the text.

L452 Why this hosing region?

The Nordic Sea is a key region connecting the Arctic and the subpolar North Atlantic. Our study aims to explore if it is possible to have an alternative relationship between the AMOC change and the Labrador Sea open-ocean deep convection change in response to an external freshwater

forcing, when the external freshwater forcing is applied over the southern Nordic Sea where there is no open-ocean deep convection.

Observational and high-resolution modelling studies suggest that the freshwater flux enters the Labrador Sea mainly through its boundary current, and the freshwater flux is largely confined within the narrow Labrador Sea boundary current with limited access to the interior Labrador Sea deep convection site (e.g., Schmidt and Send, 2007; Marsh et al., 2010; Florindo-Lopez et al., 2020; Kim et al., 2021). Hence it is less realistic to have freshwater flux released directly into the interior Labrador Sea deep convection site. Meanwhile, observational and high-resolution modelling studies suggest that the freshwater flux entering the Nordic Sea through the Fram Strait is advected southeastward into the interior southern Nordic Sea (northeast of Iceland) by the Jan Mayer Current (JMC) and the East Icelandic Current (EIC), and recirculates in the southern Nordic Sea (Miles et al. 2020; Kim et al. 2021). Paleo sea ice proxy records from the North Icelandic shelf have been employed to reconstruct the anomalous Arctic freshwater flux/sea ice exported through the Fram Strait and accumulated in the southern Nordic Sea, and they reveal multidecadal periods of substantially enhanced freshwater flux/sea ice in the southern Nordic Sea during the past several hundreds of years (Miles et al. 2020). We have added the above explanation in the Methods section.

L473-477 I agree that the representation of the boundary currents in climate models is a problem, however I disagree with this statement. Actually there is a wide range of strengths of OSNAP west in CMIP6 models (see ref 4), and it isn't obviously true that CMIP6 models generally overestimate it. It also isn't obvious true that this is related to resolution. For instance there are two resolutions of the HadGEM3 GC3.1 model in that reference, but it is the higher resolution model which has the larger Labrador sea overturning.

Thanks for the comments. We have removed the discussion about other climate models and their model resolutions here in the revision, and simply focused this sentence on the discussion of modelling biases in the boundary density in the Labrador Sea.

We agree that for the two HadGEM3 GC3.1 models, the strength of the AMOC across OSNAP West does not depend on resolution and is related more to the mean state Labrador Sea deep convection strength. We have cited ref. 4 and the two HadGEM3 GC3.1 models in the revised Introduction section.

L520 I think you need to say where it will be made available.

We have clarified in the revision that it will be made available on Zenodo, once the manuscript is published.

Reference

1. Zhang, R. (2010). Latitudinal dependence of Atlantic meridional overturning circulation (AMOC) variations. *Geophysical Research Letters*, 37(16).
2. Li, F., Lozier, M. S., Danabasoglu, G., Holliday, N. P., Kwon, Y. O., Romanou, A., ... & Zhang, R. (2019). Local and downstream relationships between Labrador Sea Water volume and North Atlantic meridional overturning circulation variability. *Journal of Climate*, 32(13), 3883-3898.
3. Zhang, R., & Thomas, M. (2021). Horizontal circulation across density surfaces contributes substantially to the long-term mean northern Atlantic Meridional Overturning Circulation. *Communications Earth & Environment*, 2(1), 112.
4. Årthun, M., Asbjørnsen, H., Chafik, L., Johnson, H. L., & Våge, K. (2023). Future strengthening of the Nordic Seas overturning circulation. *Nature Communications*, 14(1), 2065.
5. Jackson, L. C., & Petit, T. (2023). North Atlantic overturning and water mass transformation in CMIP6 models. *Climate Dynamics*, 60(9), 2871-2891.
6. Petit, T., Robson, J., Ferreira, D., & Jackson, L. C. (2023). Understanding the Sensitivity of the North Atlantic Subpolar Overturning in Different Resolution Versions of HadGEM3 - GC3. 1. *Journal of Geophysical Research: Oceans*, 128(10), e2023JC019672.
7. Tesdal, J. E., MacGilchrist, G. A., Beadling, R. L., Griffies, S. M., Krasting, J. P., & Durack, P. J. (2023). Revisiting interior water mass responses to surface forcing changes and the subsequent effects on overturning in the Southern Ocean. *Journal of Geophysical Research: Oceans*, 128(3), e2022JC019105.
8. Drake, H. F., Bailey, S., Dussin, R., Griffies, S. M., Krasting, J. P., MacGilchrist, G. A., ... & Zika, J. D. (2024). Water Mass Transformation Budgets in Finite-Volume Generalized Vertical Coordinate Ocean Models. *Authorea Preprints*.
9. Schmidt, S., & Send, U. (2007). Origin and composition of seasonal Labrador Sea freshwater. *Journal of Physical Oceanography*, 37(6), 1445-1454.
10. Marsh, R., Desbruyères, D., Bamber, J. L., De Cuevas, B. A., Coward, A. C., & Aksenov, Y. (2010). Short-term impacts of enhanced Greenland freshwater fluxes in an eddy-permitting ocean model. *Ocean Science*, 6(3), 749-760.
11. Florindo-López, C., Bacon, S., Aksenov, Y., Chafik, L., Colbourne, E., & Holliday, N. P. (2020). Arctic Ocean and Hudson Bay freshwater exports: New estimates from seven decades of hydrographic surveys on the Labrador Shelf. *Journal of Climate*, 33(20), 8849-8868.
12. Kim, W. M., Yeager, S., & Danabasoglu, G. (2021). Revisiting the causal connection between the great salinity anomaly of the 1970s and the shutdown of Labrador Sea deep convection. *Journal of Climate*, 34(2), 675-696.
13. Miles, M. W., Andresen, C. S., & Dylmer, C. V. (2020). Evidence for extreme export of Arctic sea ice leading the abrupt onset of the Little Ice Age. *Science advances*, 6(38), eaba4320.

Response to Reviewer #3

In this study, the authors analyse AMOC and Labrador Sea open-ocean convection responses in a hosing simulation. The authors analyse model output of a relatively high-resolution climate model (eddy-permitting) and impose a 0.05 Sv freshwater flux forcing at the higher latitudes (i.e., the Nordic Seas) for 80 years. The authors challenge the link between AMOC decline and a shut down of the Labrador Sea convection, they actually show the opposite response, which is interesting.

I would like to complement the authors on their detailed analysis. Higher latitudes and AMOC dynamics are complex and a lot of processes need to be considered, these processes are addressed by the authors. That being said, there is some overlap between this study and a recent study from Arthun et al. (2023). The study from Arthun et al. (2023) analyses a standard climate model (i.e., the CESM ensemble, 1° horizontal resolution) and this manuscript addresses the caveat by using a higher resolution climate model (0.25° horizontal resolution). However, Arthun et al. (2023) mention that the specific higher-resolution model GFDL-CM does not stand out from the multi-model mean of the CESM. Another difference between this study and that of Arthun et al. (2023) is the forcing, which is freshwater flux forcing and climate change (Historical and RCP8.5), respectively. In that regard, the presented study is somewhat ‘cleaner’ compared to Arthun et al. (2023).

Given all of this and theory (see below), I’m not surprised by the findings of this study. Nevertheless, the separation between AMOC strength and Labrador Sea open-ocean convection is (very) important and this is often confused by the community. The mechanisms causing the enhanced convection are also well explained and are a novel contribution. I have listed several (major) comments below, I hope that the authors find my suggestions useful.

We thank the Reviewer for the very positive and constructive comments/suggestions. We have revised the manuscript accordingly and below is our detailed response to the comments and suggestions.

Major comments and suggestions:

1. The authors convincingly demonstrate the AMOC weakening (both in depth and density coordinates) under the freshwater flux forcing. The AMOC is a 3D (4D when considering time) ocean circulation and the authors specifically analyse the AMOC strength around 60°N (the OSNAP array). They label this as ‘the AMOC’, however, this is only a minor part of the full circulation. It is known that the AMOC has different responses when comparing various latitudes (Arthun et al., 2023; Zou et al., 2020). This is missing in the current version of the manuscript, a

(supplementary) figure of the full AMOC (anomalies) under the freshwater flux forcing is needed. Such an analysis would also place the manuscript in a broader context.

We have added the new Supplementary Fig. 1 as the Reviewer suggested, which shows the AMOC anomalies under the freshwater forcing as a function of latitude. The figure demonstrates that the AMOC weakening occurs across a broad range of latitudes (extending from the subpolar region to the low latitudes), and the decline is larger across subpolar latitudes than that across low latitudes. We have also shown the AMOC weakening across the Greenland-Scotland Ridge (a maximum decline of 3.5 Sv at 28kg/m^3) in the new Supplementary Fig. 3 panel c, along with the newly added analysis of the surface water mass transformation. We have included related discussions in the revised text.

2. Deep convection \neq Sinking (e.g., Georgiou et al., 2019). The AMOC strength is linked to the sinking pathways. Deep convection is the process of vertical mixing of the water column and there is no (net) vertical displacement of water. This already suggests that AMOC responses may behave differently than Labrador Sea/Subpolar Gyre deep convection, they are likely related to some extent. It would be good to stress this more in the introduction and throughout the manuscript (e.g., Lines 111 – 114).

Thanks for the suggestion. We have stressed this point more (i.e. Deep convection \neq Sinking and that the AMOC and the Labrador Sea open-ocean deep convection involve different physical processes), and added many more related references (including Georgiou et al., 2019) along with the expanded discussion in the revised Introduction and Results sections as suggested.

3. The three different realisations show a robust result. However, it is only one climate model and the results could be model (in)sensitive (e.g., Arthun et al. 2023). I'm missing some cross-validation with other (CMIP6) models and/or observations. Observations from the OSNAP array are available, but the present-day time series is too short (< 10 years) to analyse this. The authors can compare the time mean from the OSNAP array observations and check this against their control climate model simulation. It is very difficult to conduct the exact same model set-up for various models. Alternatively, the authors could analyse a positive AMOC strength anomaly period(s) and compare this against a negative AMOC strength anomaly period(s). Such an analysis can be conducted in reanalysis products (e.g., GLORYS12V1) and (CMIP6) climate models. This places the results in a better context and whether the results are consistent against observations and/or climate models.

Thanks for the suggestions. We have added the new Supplementary Fig. 9 to compare modeled time mean AMOC streamfunctions across OSNAP West, OSNAP East, and the entire OSNAP section in our control simulation with those from the OSNAP array observations as suggested. The modeled time-mean AMOC across the entire section is dominated by that across OSNAP East,

consistent with OSNAP array observations. The modeled AMOC across OSNAP West is overestimated as discussed in the Methods section.

We analyzed the reanalysis product GLORYS12V1 from 1993 to 2023, which exhibits a weakening trend in the subpolar AMOC and an opposite strengthening trend in the Labrador Sea winter deep convection over this period. However, the AMOC variability produced in different reanalysis products often has large spread/uncertainties, hence the above opposite trends between the AMOC and the Labrador Sea deep convection over this period of the GLORYS12V1 product may not represent a robust causal relationship. Since Argo floats do not cover the deep ocean below 2000m, there are very limited observations to constrain the variability in the deep ocean (such as the variability in the overflows) below 2000m in GLORYS12V1. Hence the variability related to the deep overflows in the subpolar North Atlantic in GLORYS12V1 may not be robust.

Climate models often lack a good representation of the Iceland-Scotland overflow water (ISOW) and associated NEADW layer in the deep Labrador Sea, thus are not capable to simulate the influence of the ISOW freshening on the strengthening of the Labrador Sea open-ocean deep convection. Also, many climate models have a 1° coarse resolution, thus do not resolve well the narrow Labrador Sea boundary current that can be substantially better represented in the 0.25° model (Marsh et al., 2010). The freshwater anomalies advected by the much broader Labrador Sea boundary current in coarse resolution models would affect interior Labrador Sea unrealistically and not suitable to study the impact of upstream freshwater anomalies on downstream Labrador Sea; in contrast, the freshwater anomalies are mainly confined within the narrow Labrador Sea boundary current with limited impacts on interior Labrador Sea in the 0.25° model (Marsh et al., 2010). Hence to study the impact of the ISOW on the Labrador Sea deep convection under the external freshwater forcing, the employed climate models need to satisfy both conditions: i.e. simulating the ISOW-associated NEADW layer in the deep Labrador Sea and having at least 0.25° to represent the narrow Labrador Sea boundary current.

The observed ISOW-associated NEADW layer in the Labrador Sea is characterized by a water mass that is saltier than the Labrador Sea Water layer above (Yashayaev & Clarke, 2008). Hence there is an observed positive salinity difference between the NEADW layer and the Labrador Sea Water layer, which can be used to evaluate whether a climate model can simulate the ISOW-associated NEADW layer in the deep Labrador Sea. We have added the new Supplementary Fig. 8 to compare CMIP6 climate models for their representations of the ISOW-associated NEADW layer in the deep Labrador Sea (in term of the salinity difference between the NEADW layer and the Labrador Sea Water layer, x-axis) and their inverse horizontal grid spacing (y-axis) and search for climate models that can meet both conditions needed for the purpose of this study. This figure shows that many CMIP6 models have a negative or zero salinity difference, i.e. they cannot simulate the ISOW-associated NEADW layer in the deep Labrador Sea. Only a few models show a positive salinity difference (with GFDL CM4 and NASA-GISS having a positive salinity

difference close to the observation). Among these few models with a positive salinity difference, the majority have a 1° coarse resolution (not enough to resolve the narrow Labrador Sea boundary current), and only the model employed in this study (GFDL CM4) has a 0.25° resolution and can satisfy both conditions needed for the purpose of this study. We have added related discussions in the last paragraph of the revised Discussion section.

We have also cited observations consistent with this study in this paragraph: “The extremely strong Labrador Sea open-ocean deep convection that occurred in the early 1990s is indeed accompanied by the freshening in the upper eastern subpolar North Atlantic, the downstream ISOW, and the deep NEADW layer in the central Labrador Sea (Yashayaev and Clarke, 2008; van Aken et al., 2011), in addition to the excessive surface heat loss. The observed strengthening of the Labrador Sea ocean-ocean deep convection from the late 1960s to the early 1990s (Yashayaev and Clarke, 2008; van Aken et al. 2011) is also accompanied by the observed weakening of the ISOW over this period (Hensen et al. 2001).”

Minor comments and suggestions:

1. Line 29 – 30: The Labrador Sea/Subpolar Gyre and AMOC are different tipping elements in the climate system (Armstrong Mckay et al., 2022). This also demonstrates that AMOC changes and Labrador Sea changes are not same.

We have added a discussion of this suggested point and cited the above reference (Armstrong MaKay et al., 2022) in this paragraph in the revision.

2. Line 55: The GFDL-CM is eddy-permitting at the lower latitudes and mid-latitudes, at the higher latitudes (and continental shelves) the 0.25° horizontal resolution is too coarse. Good to add this here.

Done.

3. Line 82: I would suggest to add that the external freshwater flux forcing is applied over the entire simulation length of 80 years (as is mentioned in the Methods).

Done.

4. Line 85: Add (in brackets) the climatology value of the maximum AMOC strength.

Done.

5. Line 86 – 93: Is the AMOC weakening significant against the natural variability of the control simulation? There is no time series of the control simulation, so we cannot check this. What about any AMOC weakening/strengthening at other latitudes? (see also Major comment 1).

Yes this AMOC weakening is significant against the natural variability of the control simulation, and we have added this clarification in the revised text here. As discussed in the Methods Section,

the ‘anomaly’ in this paper refers to the ensemble-mean of the difference between the water hosing member and the corresponding control simulation segment. The ensemble mean is used throughout this study to reduce the effect of natural variability presented in each individual ensemble member and improve the signal-to-noise ratio. The residual natural variability (i.e. standard deviation) of the maximum AMOC and the AMOC at the relatively denser level (27.8kg/m^3) across the OSNAP section in the ensemble mean of the control simulation segments over the last 40 years is 0.57 Sv and 0.29 Sv respectively, both are much smaller than the amplitudes of the corresponding ensemble mean AMOC decline averaged over the last 40 years.

We have added the new Supplementary Fig. 1 to show the AMOC weakening at other latitudes, see our response to Major comment 1. The figure demonstrates that the AMOC weakening occurs across a broad range of latitudes (extending from the subpolar region to the low latitudes). We have also shown the AMOC weakening across the Greenland-Scotland Ridge in the new Supplementary Fig. 3 panel c, along with the newly added analysis/discussion of the surface and interior mixing forced water mass transformation.

6. Line 450 – 452: There is no compensation, so the global salinity content is not conserved. Good to mention it here.

Done.

7. Figure 7: Panels c and f do not have the same vertical extent as the other panels, please make consistent.

Done (Panels a, b, c have been removed to focus on the key message).

Reference

1. Georgiou, S., van der Boog, C. G., Brüggemann, N., Ypma, S. L., Pietrzak, J. D., & Katsman, C. A. (2019). On the interplay between downwelling, deep convection and mesoscale eddies in the Labrador Sea. *Ocean Modelling*, 135, 56-70.
2. Marsh, R., Desbruyères, D., Bamber, J. L., De Cuevas, B. A., Coward, A. C., & Aksenov, Y. (2010). Short-term impacts of enhanced Greenland freshwater fluxes in an eddy-permitting ocean model. *Ocean Science*, 6(3), 749-760.
3. Yashayaev, I., & Clarke, A. (2008). Evolution of North Atlantic water masses inferred from Labrador Sea salinity series. *Oceanography*, 21(1), 30-45.
4. van Aken, H. M., de Jong, M. F., & Yashayaev, I. (2011). Decadal and multi-decadal variability of Labrador Sea Water in the north-western North Atlantic Ocean derived from tracer distributions: Heat budget, ventilation, and advection. *Deep Sea Research Part I: Oceanographic Research Papers*, 58(5), 505-523.
5. Hansen, B., Turrell, W. R., & Østerhus, S. (2001). Decreasing overflow from the Nordic seas into the Atlantic Ocean through the Faroe Bank channel since 1950. *Nature*, 411(6840), 927-930.

6. Armstrong McKay, D. I., Staal, A., Abrams, J. F., Winkelmann, R., Sakschewski, B., Loriani, S., ... & Lenton, T. M. (2022). Exceeding 1.5 C global warming could trigger multiple climate tipping points. *Science*, 377(6611), eabn7950.

Response to Review Comments

Response to Reviewer #1

One minor suggestion:

The rationale of releasing freshwater in the southern Nordic Sea has been elaborated in the Methods section. Suggest including a transition sentence summarizing previous freshwater release experiments in the Labrador Sea before Line 466, and then describe their limitations.

Thank you for the suggestion. We have included a transition sentence on previous freshwater hosing experiments directly covering the interior Labrador Sea before Line 466 (i. e. before the description of their limitations) as suggested.

Response to Reviewer #2

Thank you for the manuscript “The weakening of the AMOC and associated strengthening of the Labrador Sea open-ocean deep convection”. The authors have done a good job of addressing comments in their responses, however this doesn’t really come across in the revised manuscript. Hence, there are a few major points that I will raise again – this is an issue about presentation, rather than the science, so I don’t expect that they will take long to address.

We thank the Reviewer for the further comments/suggestions. We have revised the manuscript accordingly to make it clearer and more organized.

Major

I’m still not at all clear why the authors chose this region to apply to hosing. I see that some discussion has been added, but it still doesn’t seem to say anywhere ‘we chose this region because of ...’. Is it supposed to represent some real process or targeted at something in particular, or just be a sensitivity study?

As commented by Reviewer #1, the rationale of releasing freshwater in the southern Nordic Sea has been elaborated in the Methods section (previous lines 466-479). To make it clearer, we have added a sentence in the Method section summarizing previous freshwater release experiments in which the external freshwater flux is released broadly over the entire subpolar North Atlantic and directly covers the interior Labrador Sea, followed by their limitations. At the end of our elaboration for the rationale of releasing freshwater in the southern Nordic Sea (previous lines 466-479), we have also added two sentences: “Since there is no open-ocean deep convection in the southern Nordic Sea, the anomalous freshwater flux entering this

region as discussed above will not directly affect any open-ocean deep convection. For the above reasons, we choose the southern Nordic Sea instead of the entire subpolar North Atlantic to apply the external freshwater flux”.

The authors responded with a clear answer when I asked about the main message of their paper, however there are still points in the manuscript where I got confused about what they were doing. I think this could be helped with a little reorganization and trying to bridge between topics. I would start with presenting the main results of AMOC weakening and MLD strengthening and using this to pose the main question of the paper. Then discuss the AMOC changes in more detail (including moving ‘The transient AMOC decline section’ earlier), followed, as they have done, by the tracer results and role of ISOW. I don’t think this is a major change to what they have done and means that we’re not jumping between different topics. The page or so from L138 to 176 is very confusing. There is a lot of jumping around topics. Lines 138-156 seem very much related to the transient AMOC discussion, so I suggest that they are removed or amalgamated (see also next point). The authors also need to point the author to why they are doing something when they change to a different topic. This is done in some of the later sections but not in the first two subsections of the results.

Thanks for the comments. As suggested, we have reorganized the text, deleted most of Lines 140-156, and replaced them with ‘The transient AMOC decline across OSNAP East’ subsection, which has been moved earlier. We have also added more motivation/transition sentences for different subsections of the results to clarify our purpose.

I still have concerns around the analysis of density gradients in lines 138-156. In their previous response to my questions about this, the authors said that they were not relying on the usual relationship between depth space AMOC and boundary density/pressures, but instead on a relationship between density space AMOC and density gradients from another paper. Firstly, this isn’t clear in the revised text – instead the authors refer generally to the relationship between the AMOC and densities, referring to a number of papers, rather than making it clear that they are not relying on the usual arguments. Secondly, the papers they refer to (Zhang and Thomas 2021 and Arthun et al 2023) do not demonstrate a definitive relationship between the AMOC in density space and density gradients. I certainly couldn’t see anything to suggest that the AMOC at a specific deep density levels was related to density changes in specific regions. Although the AMOC change in density space is quite likely related to density changes, it isn’t clear which regions are important for a given ‘relatively denser’ layer. We know that depth and density space AMOC are different at these latitudes, so they can’t both have a simple relationship to zonal density gradients. To be honest I’d delete this paragraph and bring the ‘Transient AMOC decline’ subsection forwards.

Thanks for the suggestion. We have deleted most of lines 138-156 for simplicity and replaced them with the ‘Transient AMOC decline across OSNAP East’ subsection, which has been moved forward as suggested.

If there are no zonal density gradients, then both the depth-space and density-space AMOC would disappear. Both the depth-space and density-space AMOC are related to the zonal density gradients, but they are related in different complicated ways. We interpret the west-east density contrast as a simplified indicator for the transient density-space AMOC changes at the relatively dense level (because the zonal density contrast at a specific density level affects the contribution of the horizontal gyre across sloping isopycnals to the density-space AMOC at that density level).

Minor

L15 and throughout. The author's keep using the term 'the relatively denser level' and similar throughout the paper. I'm not a grammar expert, but this doesn't sound right to me (I'd say 'denser level' or 'relatively dense level'). There's also the issue that it's rather vague – it's difficult to tell whether this is a given level that is referred to throughout (in which case it might be best to define it as a symbol like ρ_0), or just that it's 'relatively' dense (in comparison to what?).

We have revised the term to 'the relatively dense level', and we have used 'the relatively dense level around $\sigma_0 = 27.84 \text{ kg m}^{-3}$ ' to make it clear that it refers to a given level around $\sigma_0 = 27.84 \text{ kg m}^{-3}$ throughout. We have also added the clarification that it is 'a level denser than the level of the maximum AMOC' when this term is mentioned for the first time in the text.

L107 How does the AMOC weakening compare to the variability in the control? The same goes for the MLD strengthening – is it significant compared to variability, or is it just that it doesn't weaken?

Yes, both the AMOC weakening and the Labrador Sea MLD strengthening are statistically significant against the residual natural variability in the control ensemble, and we have added this clarification in the revised text.

L109 What is meant here by significant? Is there a statistical significance test?

Yes, we did the statistical significance test using the 95% confidence level and we mean 'statistically significant' here. We have changed 'significant' to 'statistically significant' to make it clear in the revised text.

L131,134 'due' and 'caused' – Given the indirect relationship between the AMOC and the LS convection, we wouldn't expect one to cause the other. Maybe you want to say they are not related or similar?

We have removed 'due' and 'caused' here and pointed out that the AMOC and the LS convection changes are opposite.

L160 'Results are also consistent with observational analyses that suggest that the inferred historical weak AMOC phase is associated with...'

Done

L162-176. You need to introduce what the WMT is first and what you're wanting to do with the analysis. Also reference to the methods for calculating it.

We have added that "The Water mass transformation (WMT) refers to the diabatic processes by which water masses transform from one density class to another, influencing/balancing the large-scale ocean circulation such as the AMOC (Walín, 1982). The WMT includes the surface forced (WMT_S) and the interior mixing forced (WMT_M) components (see Methods section for the calculation methods). Our analysis investigates how the long-term mean surface and interior mixing forced water mass transformation (WMT_S and WMT_M) changes influence/balance the long-term mean density-space AMOC decline." We have also cited the methods for calculating the WMT in the Methods section.

L215 upper Nordic Sea

Done

L219-228 This is certainly not obvious from what is shown. Fig 6 doesn't show the Nordic seas, so we can't see how much dye there is there going over the overflows. Also could there not be sinking/mixing from the upper to lower waters along the boundary of the Irminger and Labrador seas?

We have added Supplementary Fig. 6, which is similar to Fig. 5 but is extended to 70°N to include the southern Nordic Sea and at a deeper level (625m) to better show that the dye is carried out of the Nordic Sea by the overflows across the Greenland-Scotland Ridge.

There is sinking/mixing from the upper to lower waters along the boundary of the Irminger and Labrador seas. We have clarified this information by revising the sentence right before: "The upper ocean dye propagates along the western boundary of the Irminger and Labrador seas with partial sinking/mixing into the deeper ocean, and is partially mixed into the interior ocean including the Labrador Sea open-ocean deep convection region (Fig. 6, 7, Supplementary Fig. 6)".

L291-294 I don't think this is necessarily true. There is also a signal from dye originating from the Denmark Straits, but also it's not obvious that it couldn't partly come from the upper LS as well (see previous comment).

The absolute freshening could partly come from the upper LS as well or from the mixing with the signal along the western boundary, and we have discussed that in our response to the previous comment. However, here we focus on the relative enhanced freshening, i. e., the stronger freshening in the deep NEADW layer (a layer of water that originates from the Iceland-Scotland overflow) in the deep central LS than the freshening in the upper central LS. To avoid any confusion, we have deleted this sentence here and replaced it with the following clarification: "The freshening in the NEADW layer in the deep central Labrador Sea is stronger than that in the upper central Labrador Sea (Fig. 3d, Supplementary Fig.7a)". The analysis of the relative enhanced freshening in the NEADW layer is discussed in detail in the following paragraph on the difference between the salt-based and the dye-based FWF

anomalies ($FWF'_{salt} - FWF'_{dye}$).

L340-344 Is this shown? Please add reference to figure or 'not shown' if you didn't include it.

Done

L373-385 These two paragraphs seem to be very much stuck on the end. I'd suggest you find more relevant places to put each one.

We have moved the first of these two paragraphs to the more relevant Methods section, and added a transition sentence for the second of these paragraphs, so that it is more relevant and better connected with other discussions of this subsection.

L373-379 It would be interesting to include density anomalies of upper and lower densities in Supp Fig 7. Is it true that the variability is coming from the upper layer and trend from the lower layer or is it more complex?

We have added Supplementary Fig. 7b of the surface and deep ocean density anomalies in the Labrador Sea as suggested. Yes, it is true that the variability is coming from the upper layer and the strengthening trend in the Labrador Sea March MLD is caused by a stronger decrease in the lower layer density after 40 years.

L380-385 I think the last sentence could be expanded on a bit, including reference to figures. Is this freshening seen in the tracer for example? Eg is it directly from the forcing?

This freshening can be seen in the upper ocean salt-based and dye-based FWF anomalies (Fig.6). We have included the reference to Fig. 6 and the above clarification in the revised last sentence here.

L427 "... in the deep Labrador Sea and boundary currents, both of which are important to represent this process" Or similar

Done

L431-433 However 0.25 degree models are not able to simulate eddies which mix between the boundary and interior LS. So this is a caveat for whether this model is properly representing this process.

Here this sentence summarizes previous modeling results. We have added another citation (Kim et al. 2021) here, which shows that in the 0.1 degree model, the freshwater anomalies are also confined within the substantially better-resolved narrow Labrador Sea boundary current with limited impact on interior Labrador Sea. We have also added in the end of this paragraph that "The 0.25° models are still not able to fully resolve eddies, and higher-resolution models would be needed in future studies to provide a more accurate representation of the eddy mixing process between the boundary and the interior Labrador Sea".

L483-4 There is quite a lot of repetition.

We have removed the repetition here.

L506 Normally I think of stratification as being from the surface layer. You should note that this is deep stratification.

We have clarified that this is deep stratification.

Response to Reviewer #3

I would like to thank the authors for their replies to all the reviewers. The clarifications, additional analyses and results helped to strengthen the main messages in the revised manuscript. I would also like to thank the authors for their analysis on the CMIP6 data, I know that it can be quite a challenge in analysing the different CMIP6 models. These CMIP6 results prompt some further questions, but they are beyond the scope of this manuscript.

I'm happy to recommend the manuscript for publication.

Thanks for the recommendation.

Review of “The weakening of the AMOC and associated strengthening of the Labrador Sea open-ocean deep convection” by Wei and Zhang.

Summary

The study investigates the responses of AMOC and Labrador Sea convection to external freshwater forcing applied in the southern Nordic Sea in an eddy-permitting coupled climate model. They found a decline of the AMOC associated with reduced zonal density contrast across the subpolar North Atlantic, which was further attributed to both direct impact from freshwater anomalies and indirect contributions from circulation/mixing changes. In contrast to a weakening AMOC, they found a strengthening convection in the Labrador Sea. The enhanced convection was further explained by decreased density (enhanced freshening) in the overflow water layer that reduced the vertical stratification. These results suggest that Labrador Sea convection is not the cause of the AMOC decline.

I find the study very interesting and unique. The finding that the AMOC and Labrador Sea convection have opposite trends under external freshwater forcing is novel, and the conclusion that weakened AMOC leads to enhanced convection by reducing vertical stratification is noteworthy. The manuscript is overall well-written, and the presentations of the results are clear. That being said, I do have several concerns in terms of the freshwater release experiment design and the model results validations, which I believe could be improved by further discussions and analyses.

Major comments

(1). The rationale of the freshwater release location needs to be elaborated. Results of the freshwater hosing experiments may depend on where the external freshwater flux is injected. Previous studies uniformly inject freshwater over the Labrador Sea and thus find a causal relationship between convection and AMOC. The current study, on the other hand, releases freshwater in the southern Nordic Sea and finds opposite trends of AMOC and Labrador Sea convection. Which freshwater release site is more realistic? Is there observational evidence that supports enhanced freshwater flux over southern Nordic Sea?

(2). The Labrador Sea stratification or mixed layer depth in control simulation needs to be validated with observations. The conclusion that overflow layer density impacts convection by modifying vertical stratification is valid when convection frequently reaches the deep overflow layer. However, observed convection in the Labrador Sea is ~1000-

1500m (except for that in the early 1990s), which is above the overflow layer and is less influenced by the overflow density change. If the model overestimates convection depth, the impact from overflow layer density could very likely be exaggerated.

(3). Further discussion on the relationship between Labrador Sea convection and the AMOC is suggested. As stated in the manuscript, the OSNAP observations have revealed that overturning in the eastern basin dominates the subpolar AMOC. In the current study, it is further shown that AMOC decline in response to freshwater forcing is determined by circulation and property changes in the eastern subpolar basin. How to reconcile these results with previous studies that suggest the AMOC variability determined by Labrador Sea convection? Is it a matter of time scales? I think the relationship between convection and the AMOC worth in-depth discussion.

Minor comments

(1). Please indicate in the title that the changes are in response to external freshwater flux.

(2). Lines 146-158: It is a bit difficult to tell horizontal circulation change just by looking at Fig.2. Reporting magnitudes of AMOC change in density and depth space would help.

(3). Line 163: I would suggest using abyssal circulation instead of recirculation.

(4). Line 195: I am confused by how freshwater anomalies get into the deep ocean. I thought the fresh anomalies would mostly stay in the near-surface layer, and most of the mixing and entrainment occurred in the bottom layer. In other words, the freshwater anomalies cannot be directly entrained into the deep layer.

(5). Fig. 1: There is an interesting decadal variability of AMOC at OSNAP-West, which seems to be associated with Labrador Sea MLD decadal variability. Is this relationship present in the control simulation?

In this study, the authors analyse AMOC and Labrador Sea open-ocean convection responses in a hosing simulation. The authors analyse model output of a relatively high-resolution climate model (eddy-permitting) and impose a 0.05 Sv freshwater flux forcing at the higher latitudes (i.e., the Nordic Seas) for 80 years. The authors challenge the link between AMOC decline and a shut down of the Labrador Sea convection, they actually show the opposite response, which is interesting.

I would like to complement the authors on their detailed analysis. Higher latitudes and AMOC dynamics are complex and a lot of processes need to be considered, these processes are addressed by the authors. That being said, there is some overlap between this study and a recent study from Arthun et al. (2023). The study from Arthun et al. (2023) analyses a standard climate model (i.e., the CESM ensemble, 1° horizontal resolution) and this manuscript addresses the caveat by using a higher resolution climate model (0.25° horizontal resolution). However, Arthun et al. (2023) mention that the specific higher-resolution model GFDL-CM does not stand out from the multi-model mean of the CESM. Another difference between this study and that of Arthun et al. (2023) is the forcing, which is freshwater flux forcing and climate change (Historical and RCP8.5), respectively. In that regard, the presented study is somewhat ‘cleaner’ compared to Arthun et al. (2023).

Given all of this and theory (see below), I’m not surprised by the findings of this study. Nevertheless, the separation between AMOC strength and Labrador Sea open-ocean convection is (very) important and this is often confused by the community. The mechanisms causing the enhanced convection are also well explained and are a novel contribution. I have listed several (major) comments below, I hope that the authors find my suggestions useful.

Major comments and suggestions:

1. The authors convincingly demonstrate the AMOC weakening (both in depth and density coordinates) under the freshwater flux forcing. The AMOC is a 3D (4D when considering time) ocean circulation and the authors specifically analyse the AMOC strength around 60°N (the OSNAP array). They label this as ‘the AMOC’, however, this is only a minor part of the full circulation. It is known that the AMOC has different responses when comparing various latitudes (Arthun et al., 2023; Zou et al., 2020). This is missing in the current version of the manuscript, a (supplementary) figure of the full AMOC (anomalies) under the freshwater flux forcing is needed. Such an analysis would also place the manuscript in a broader context.
2. Deep convection \neq Sinking (e.g., Georgiou et al., 2019). The AMOC strength is linked to the sinking pathways. Deep convection is the process of vertical mixing of the water column and there is no (net) vertical displacement of water. This already suggests that AMOC responses may behave differently than Labrador Sea/Subpolar Gyre deep convection, they are likely related to some extent. It would be good to stress this more in the introduction and throughout the manuscript (e.g., Lines 111 – 114).
3. The three different realisations show a robust result. However, it is only one climate model and the results could be model (in)sensitive (e.g., Arthun et al. 2023). I’m missing some cross-validation with other (CMIP6) models and/or observations. Observations from the OSNAP array are available, but the present-day time series is too short (< 10 years) to analyse this. The authors can compare the time mean from the OSNAP array observations and check this against their control climate model simulation. It is very difficult to conduct the exact same model set-up for various models. Alternatively, the authors could analyse a positive AMOC strength anomaly period(s) and compare this against a negative AMOC strength anomaly period(s). Such an analysis can be conducted in reanalysis products (e.g., GLORYS12V1) and (CMIP6) climate models. This places the results in a better context and whether the results are consistent against observations and/or climate models.

Minor comments and suggestions:

1. Line 29 – 30: The Labrador Sea/Subpolar Gyre and AMOC are different tipping elements in the climate system (Armstrong McKay et al., 2022). This also demonstrates that AMOC changes and Labrador Sea changes are not same.
2. Line 55: The GFDL-CM is eddy-permitting at the lower latitudes and mid-latitudes, at the higher latitudes (and continental shelves) the 0.25° horizontal resolution is too coarse. Good to add this here.
3. Line 82: I would suggest to add that the external freshwater flux forcing is applied over the entire simulation length of 80 years (as is mentioned in the Methods).
4. Line 85: Add (in brackets) the climatology value of the maximum AMOC strength.
5. Line 86 – 93: Is the AMOC weakening significant against the natural variability of the control simulation? There is no time series of the control simulation, so we cannot check this. What about any AMOC weakening/strengthening at other latitudes? (see also Major comment 1).
6. Line 450 – 452: There is no compensation, so the global salinity content is not conserved. Good to mention it here.
7. Figure 7: Panels c and f do not have the same vertical extent as the other panels, please make consistent.

References:

1. Armstrong McKay et al. (2022), <https://www.science.org/doi/10.1126/science.abn7950>
2. Arthun et al. (2023), <https://doi.org/10.1038/s41467-023-37846-6>
3. Georgiou et al. (2019), <https://doi.org/10.1016/j.ocemod.2019.02.004>
4. Zou et al. (2020), <https://doi.org/10.1175/JCLI-D-19-0215.1>